# Spatially-Adaptive Gradient Re-parameterization for 3D Large Kernel Optimization

Ho Hin Lee[1]   Quan Liu[2]   Shunxing Bao[1]   Yuankai Huo[1]   Bennett A. Landman[1]

## Abstract

Large kernel convolutions offer a scalable alternative to vision transformers for high-resolution 3D volumetric analysis, yet naively increasing kernel size often leads to optimization instability. Motivated by the spatial bias inherent in effective receptive fields (ERFs), we theoretically demonstrate that structurally re-parameterized blocks induce spatially varying learning rates that are crucial for convergence. Leveraging this insight, we introduce Rep3D, a framework that employs a lightweight modulation network to generate receptive-biased scaling masks, adaptively re-weighting kernel updates within a plain encoder architecture. This approach unifies spatial inductive bias with optimization-aware learning, avoiding the complexity of multi-branch designs while ensuring robust local-to-global convergence. Extensive evaluations on five 3D segmentation benchmarks demonstrate that Rep3D consistently outperforms state-of-the-art transformer and fixed-prior baselines. The source code is publicly available at https://github.com/leeh43/Rep3D.

## 1. Introduction

The landscape of medical vision models has evolved rapidly, expanding from early convolutional architectures to modern transformer-based designs. In particular, Vision Transformers (ViTs) have gained traction for their ability to model long-range dependencies using multi-head self-attention and minimal inductive bias (Dosovitskiy et al., 2020). In parallel, the community has revisited large kernel convolutions as a scalable alternative to attention mechanisms, particularly in the context of high-resolution 3D volumetric data (Liu et al., 2022b; Lee et al., 2022). Despite architectural differences, both ViTs and large-kernel CNNs share a central goal: expanding the effective receptive fields (ERFs) to enable rich spatial context aggregation. However, simply increasing kernel size does not guarantee improved performance. Prior work has shown that naive enlargement of convolutional filters can result in saturated or degraded accuracy across various segmentation tasks (Ding et al., 2022b; Lee et al., 2023). Unlike ViTs, which adaptively attend to spatial content, standard convolutions rely on static, weight-shared kernels and lack the ability to modulate importance across spatial positions. This limitation prompts our first research question: **Can we incorporate spatial priors into large kernel convolutions to improve learning effectiveness?**

Recent advances in structural re-parameterization offer a promising direction. Methods such as RepLKNet (Ding et al., 2022b), SLaK (Liu et al., 2022a), and PELK (Chen et al., 2024) scale kernels to extreme sizes (e.g., $31 \times 31$, $51 \times 51$, $101 \times 101$) by combining parallel branches of "large + small" convolutions into what is referred to as a Constant-Scale Linear Addition (CSLA) block. These parallel paths are merged into a single kernel at inference time, enabling efficient deployment while capturing multi-scale features during training. Interestingly, we observe that CSLA blocks naturally encode spatial learning bias: elements near the kernel center tend to converge faster than those on the periphery. This mirrors diffusion-like gradient propagation in ERFs starting from the center and expanding outward. These observations suggest that convergence dynamics are not uniform across the kernel, but instead spatially structured. This leads to our second question: **Can we explicitly model this diffusion pattern as a learnable spatial prior to re-weight kernel element updates during training?**

To address this, we first provide a theoretical analysis of the optimization dynamics in CSLA-based re-parameterized convolutions. We show that each branch (e.g., small vs. large kernels) implicitly operates under a distinct learning rate, leading to element-wise differences in convergence speed. These dynamics correlate with ERF visualizations and share characteristics with spatial frequency patterns in human visual perception (Kulikowski et al., 1982). Inspired by this, we propose a novel receptive bias

[1]Department of Electrical and Computer Engineering, Vanderbilt University, Nashville, USA [2]Accenture, USA. Correspondence to: Ho Hin Lee <ho.hin.lee@vanderbilt.edu>.

*Proceedings of the 43rd International Conference on Machine Learning*, Seoul, South Korea. PMLR 306, 2026. Copyright 2026 by the author(s).

re-parameterization strategy that encodes spatial distance from the kernel center as a spatial bias prior on learning convergence. As shown in Figure 1, we implement this as a low-rank modulation mechanism that generates spatial scaling factors for kernel weights, allowing the optimizer to emphasize local versus global regions adaptively for gradient back-propagation. Building on this insight, we present Rep3D, a 3D convolutional architecture that integrates large kernel convolutions (e.g., $21 \times 21 \times 21$) with our proposed re-parameterization approach. Unlike prior approaches that rely on multi-branch structures, Rep3D employs a plain and efficient encoder to reduce complexity while preserving representational capacity. We evaluate Rep3D across five challenging volumetric medical segmentation benchmarks and show that it consistently outperforms state-of-the-art transformer- and CNN-based models. Our key contributions are as follows:

- We propose Rep3D, a 3D CNN with large kernel convolutions and a streamlined encoder design that achieves state-of-the-art (SOTA) performance on multi-scale (i.e. from organs/tissues to tumors) segmentation benchmarks.

- We propose a novel and theoretically grounded re-parameterization approach that models ERF diffusion as a learnable spatial bias prior, enabling element-wise modulation of gradient convergence for training.

- We validate our method on five challenging 3D medical imaging benchmarks under direct training settings, achieving consistent and significant improvements across all datasets.

## 2. Related Work

**CNN-based 3D Models:** Foundational architectures such as 3D U-Net (Çiçek et al., 2016), V-Net (Milletari et al., 2016), and nnU-Net (Isensee et al., 2021) have played a pivotal role in establishing the standard for volumetric medical image segmentation. These models rely on encoder-decoder designs with dense skip connections, offering a strong balance between spatial resolution and semantic representation. Due to their stability, interpretability, and effectiveness without requiring large-scale pre-training, they remain widely used in clinical and research benchmarks. However, their inherently local receptive fields limit their ability to capture long-range dependencies, motivating subsequent architectural innovations aimed at expanding the effective receptive field (ERF).

**Vision Transformer-based and Hybrid Models:** To overcome the locality constraints of CNNs, transformer-based architectures such as UNETR (Hatamizadeh et al., 2022b) and SwinUNETR (Hatamizadeh et al., 2022a) introduce global self-attention mechanisms that model distant spatial dependencies more effectively (e.g., follow-up models like nnFormer (Zhou et al., 2021), Swin-Unet (Cao et al., 2021), and SwinBTS (Jiang et al., 2022)). These models encode context across entire volumes through hierarchical token representations, marking a major shift in design philosophy. However, they typically require large-scale pre-training and introduce significant computational complexity due to the quadratic scaling of attention, particularly problematic in 3D volumetric settings. Additionally, their reliance on patch-based tokenization can compromise fine-grained spatial precision—crucial in dense prediction tasks like medical segmentation.

**Large Kernel Convolution Networks:** A more recent and efficient alternative to transformers involves expanding the ERF through large kernel convolutions, as demonstrated by models such as ConvNeXt (Liu et al., 2022b), 3D UX-Net (Lee et al., 2022), and MedNeXt (Roy et al., 2023). These architectures leverage depth-wise or separable convolutions to approximate global context modeling while preserving the simplicity and inductive biases of convolutional designs. However, studies in 2D vision backbones (e.g., RepLKNet (Ding et al., 2022b) (kernel size: $31 \times 31$), SLaK (Liu et al., 2022a) (kernel size: $51 \times 51$)) reveal that naively scaling up kernel size leads to saturation or performance degradation in the absence of additional structural guidance. This key insight motivates the design of Rep3D, which augments large 3D kernels with a learnable spatial prior inspired by ERF theory. By explicitly guiding convergence dynamics across kernel elements, Rep3D enables more effective utilization of large kernels, bridging the gap between CNN efficiency and transformer-like contextual modeling.

**The Integration of Weight Re-parameterization.** Structural re-parameterization (SR) has emerged as a powerful paradigm to enhance CNN training without altering inference-time complexity. Models like RepVGG (Ding et al., 2021) and OREPA (Hu et al., 2022) employ additional convolution branches (e.g., $1 \times 1$ or identity paths) during training to improve gradient flow and feature diversity. These branches are merged into a single convolution kernel post-training, allowing for efficient inference. RepLKNet (Ding et al., 2022b) and SLaK (Liu et al., 2022a) extend this approach to large 2D kernels, increasing the receptive field while maintaining tractable inference cost via kernel decomposition or sparse groups. A complementary line of work focuses on gradient re-parameterization instead of modifying model weights directly. RepOptimizer (Ding et al., 2022a), for example, modifies the back-propagation process by applying learnable scaling to gradient updates, enabling effective training of plain CNNs. While much of the re-parameterization research has focused on 2D natural images, extending these methods to 3D medical imaging presents unique challenges. Volumetric kernels require

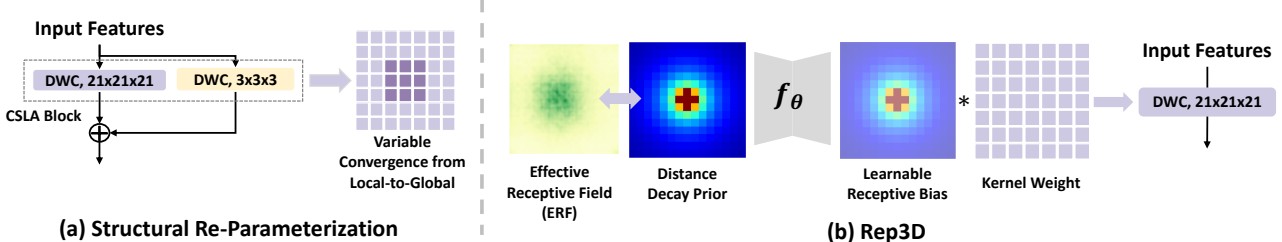

*Figure 1.* (a) Traditional structural re-parameterization methods (e.g., CSLA blocks) re-parameterize small and large kernel convolutions to improve representational capacity, but apply linear optimization with the same learning rate across the kernels, demonstrating faster convergence in local regions than global ones. (b) In contrast, Rep3D introduces a learnable spatial bias via a generator network $f_\theta$, which modulates each element in the large kernel using a prior based on distance decay. This adaptive modulation enables local-to-global update dynamics aligned with ERF behavior, enhancing both training stability and model performance for 3D volumetric tasks.

significantly more parameters, and naïve kernel expansion leads to high computational costs and optimization instability. 3D RepUX-Net (Lee et al., 2023) demonstrated an initial attempt to adapt weight re-parameterization to 3D medical imaging and scale large depthwise kernels with fixed prior context, but it still relies on a hand-designed spatial prior and lacks flexibility in learning structured optimization biases over 3D kernel locations. To bridge this gap, there is growing interest in using spatial priors or effective receptive field modeling to guide re-parameterization for large kernel learning in the 3D setting.

## 3. Rep3D

Unlike RepOptimizer, which re-parameterizes gradient magnitudes globally through learnable scaling, Rep3D rethinks the training dynamics of large-kernel convolution by explicitly embedding spatial bias, derived from effective receptive fields (ERFs), into the optimization process. Motivated by structural re-parameterization (SR) and the distinctive gradient behavior observed in ERFs, Rep3D introduces a low-rank, learnable re-parameterization that adapts element-wise update behavior across the kernel. We first derive the theoretical equivalence between parallel convolution branches and their single-operator counterparts, showing that a "large + small" convolution block (as in RepLKNet (Ding et al., 2022b)) implicitly assigns spatially varying learning rates. We then translate this insight into a unified formulation and construct a lightweight generator that outputs a convergence-aware modulation mask, as shown in Figure 2. The output modulated mask models fine-grained learning dynamics during training, improving both scalability and performance in 3D tasks with large kernel convolution.

### 3.1. Spatially Non-Uniform Optimization in Parallel Branches

Structural re-parameterization improves the optimization of large-kernel convolutions by training a multi-branch block and merging it into a single convolution at inference. In

this section, we revisit this mechanism from the perspective of kernel-element optimization. Our goal is not to prove that structural re-parameterization alone explains all empirical gains, but to expose a simple and useful mechanism: parallel large- and small-kernel branches induce spatially non-uniform update dynamics over the equivalent large kernel. Consider a two-branch convolutional block consisting of a large 3D kernel $W_L \in \mathbb{R}^{K_L \times K_L \times K_L}$ and a smaller 3D kernel $W_S \in \mathbb{R}^{K_S \times K_S \times K_S}$, where $K_S < K_L$. The block output is

$$Y = \alpha_L(X * W_L) + \alpha_S(X * W_S), \qquad (1)$$

where $\alpha_L$ and $\alpha_S$ are branch scaling factors. Since convolution is linear in the kernel, the two branches can be merged into an equivalent large kernel,

$$W' = \alpha_L W_L + \alpha_S \mathrm{Pad}(W_S), \qquad (2)$$

where $\mathrm{Pad}(\cdot)$ embeds the small kernel into the center of the large-kernel support. This equivalent-kernel view is commonly used to obtain a plain single-branch convolution at inference. Here, we use it to analyze how the training-time branches affect the update pattern of different spatial positions in the merged kernel. Let $\Omega_L$ denote the full spatial support of the large kernel and $\Omega_S \subset \Omega_L$ denote the central support covered by the padded small kernel. During training, the large branch updates all positions in $\Omega_L$, while the small branch only updates positions in $\Omega_S$. Therefore, a kernel element at spatial offset $\mathbf{p} \in \Omega_L$ receives different effective training signals depending on whether it lies inside the small-kernel region. Abstractly, the update of the equivalent kernel can be written as:

$$\Delta W'(\mathbf{p}) = \begin{cases} \alpha_L \Delta W_L(\mathbf{p}) + \alpha_S \Delta W_S(\mathbf{p}), & \mathbf{p} \in \Omega_S \\ \alpha_L \Delta W_L(\mathbf{p}), & \mathbf{p} \in \Omega_L \setminus \Omega_S \end{cases}$$
$$(3)$$

Eq. 3 describes the branch-wise parameter update of the equivalent kernel. To isolate the spatial optimization bias induced by this update, we abstract away the gradient directions and summarize the branch contributions as an effective learning strength. Under first-order optimization with

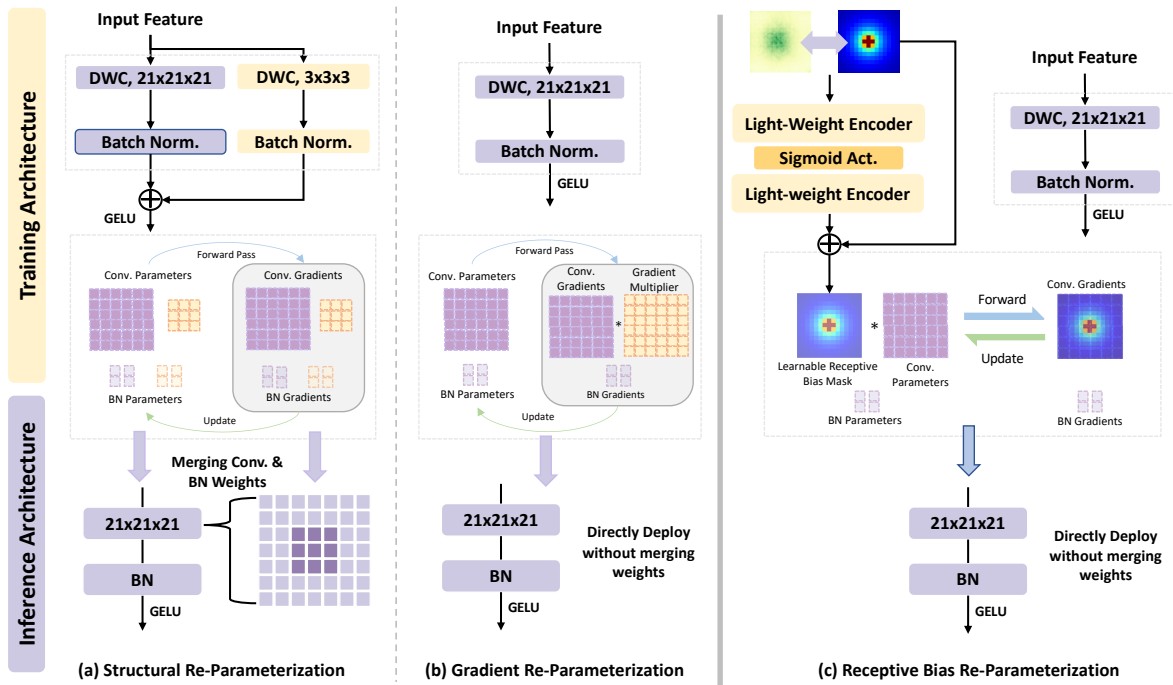

*Figure 2.* In contrast to (a) structural or (b) gradient-based re-parameterization, Rep3D introduces a novel re-parameterization strategy that injects a learnable spatial bias into large kernel convolutions for optimization. During training, a lightweight generator network produces a modulation mask conditioned on a distance-based prior, which adaptively scales gradient updates across the kernel. This enables spatially-aware learning dynamics that reflect local-to-global variations in the effective receptive field (ERF).

branch learning rates $\eta_L$ and $\eta_S$, this yields:

$$\lambda_{\text{eff}}(\mathbf{p}) = \begin{cases} \alpha_L \eta_L + \alpha_S \eta_S, & \mathbf{p} \in \Omega_S \\ \alpha_L \eta_L, & \mathbf{p} \in \Omega_L \setminus \Omega_S \end{cases} \quad (4)$$

where $\lambda_{\text{eff}}$ is the effective element-wise learning rate inherited from the two branch-specific updates. Equation 4 shows that the center region of the equivalent kernel receives a stronger effective update than the peripheral region, because central elements are optimized through both the large and small branches. In contrast, peripheral elements are updated only through the large branch. Thus, even though the final inference model is a single large convolution, the training-time parameterization imposes a spatially non-uniform optimization bias over the kernel support. This perspective provides a useful interpretation of why large-kernel structural re-parameterization can improve optimization. The auxiliary small branch implicitly emphasizes the central kernel region, which is consistent with the effective receptive field phenomenon: central offsets tend to dominate the contribution to the output, while distant offsets are harder to optimize and often contribute less during early training. However, this bias is fixed by the hand-designed small-kernel support. Once $K_S$ is chosen, the region receiving stronger updates is predetermined and cannot adapt to the spatial statistics of 3D medical anatomy. This limitation

motivates Rep3D. Rather than relying on an implicit center-only update bias induced by a fixed auxiliary branch, Rep3D learns a structured spatial modulation field over the full 3D kernel support. This modulation field allows different kernel locations to receive different optimization strengths during training, while the resulting model can still be merged into a plain large-kernel convolution for inference. We provide the full branch-wise derivation of the equivalent-kernel update in Appendix A.

### 3.2. Low-Rank Receptive Bias Modeling (LRBM)

As the above theory validates the correlation between variable learning and local-to-global gradient dynamics in ERF, we argue that such receptive bias can enhance the efficiency of learning large convolution kernels. We model the diffusion behavior of ERF with a reciprocal distance decay function $f_d$ and generate a prior mapping $P \in \mathbb{R}^{C \times 1 \times K \times K \times K}$ for weight re-parameterization as follows:

$$f_d(x, y, z, c) = \sqrt{(x-c)^2 + (y-c)^2 + (z-c)^2}, \quad (5)$$

$$P = \frac{\beta}{f_d(x_k, y_k, z_k, c) + \beta}, \quad (6)$$

where $k$ and $c$ are the element and central index of the kernel weight, and $\beta$ is a learnable parameter initialized

to 0 that controls the weight distribution of the distance mapping. However, such a fixed prior mapping lacks the flexibility to learn spatially structured optimization biases beyond a hand-designed distance prior. To address this, we propose to adapt learnable spatial bias by co-training a lightweight 2-layer generator network $f_\theta : \mathbb{R}^{C \times 1 \times K \times K \times K} \to \mathbb{R}^{C \times 1 \times K \times K \times K}$. We generate an adaptive mask $M$ for depthwise convolution kernels with low computation cost as follows:

$$M = P + f_\theta(P), \tag{7}$$

with the generator defined as:

$$f_\theta(P) = \text{Norm}_2\Big(\text{DConv}_2\big(\sigma(\text{Norm}_1(\text{DConv}_1(P)))\big)\Big), \tag{8}$$

where $\text{DConv}_1$ and $\text{DConv}_2$ are 3D depthwise convolutions with kernel size 7 and padding 3, $\text{Norm}_1$ and $\text{Norm}_2$ are layer normalizations, and $\sigma$ is a sigmoid activation ensuring all scaling values are between 0 and 1. Such a learnable function captures a structured spatial weighting pattern over kernel locations while preserving computational efficiency. The resulting modulation mask $M$ is then used to re-parameterize the kernel weights:

$$W_{\text{eff}} = W \odot M, \tag{9}$$

where $W$ is the original convolution kernel and $\odot$ denotes element-wise multiplication. Importantly, the mask is applied during training only, and the learned generator can be removed during inference for efficiency.

### 3.3. Network Architecture

The overall network architecture to validate Rep3D builds upon the encoder–decoder structure of 3D UX-Net (Lee et al., 2022), which processes volumetric data through hierarchical resolution stages with skip connections to preserve fine-grained spatial features. Unlike prior transformer-based models or heavily modular CNNs, our design favors plain convolution blocks to minimize computational burden while preserving capacity for large-scale context modeling. Following insights from prior work (Ding et al., 2022b), we adopt a $21 \times 21 \times 21$ depthwise convolution (DWC-21) as the kernel backbone, which we empirically identify as the best trade-off between expressiveness and efficiency in 3D. Each encoder block consists of batch normalization, followed the depthwise convolution and GELU activation. The feature propagation from layer $\ell - 1$ to layer $\ell$ and then to $\ell + 1$ is defined as:

$$\hat{z}_\ell = \text{GELU}\big(\text{DWC}_{21}(\text{BN}(z_{\ell-1}))\big), \tag{10}$$

$$\hat{z}_{\ell+1} = \text{GELU}\big(\text{DWC}_{21}(\text{BN}(\hat{z}_\ell))\big), \tag{11}$$

where $z_{\ell-1}$ is the input from the previous layer, $\hat{z}_\ell$ and $\hat{z}_{\ell+1}$ are intermediate representations, BN denotes batch normalization, and $\text{DWC}_{21}$ represents depthwise convolution with

a $21^3$ kernel. This architectural choice allows the network to efficiently encode both local and global context, while enabling seamless integration of our re-parameterized learning framework.

## 4. Experimental Setup

**Datasets and Implementation Details.** We evaluate Rep3D on five publicly available volumetric segmentation datasets, covering a wide range of anatomical structures across different spatial scales, from large organs (e.g., liver, stomach) to smaller and more challenging targets (e.g., tumors, vessels). We report results using the Dice Similarity Coefficient (DSC) as the primary evaluation metric, quantifying spatial overlap between predicted segmentations and ground truth labels. Additional details, including dataset resolution normalization, voxel spacing, and pre-processing pipelines and experimental details are provided in the appendix.

## 5. Results

### 5.1. Evaluation on Tissue & Tumor Segmentation

To assess the generalization and scalability of Rep3D across diverse anatomical structures and clinical targets, we evaluate performance on three representative volumetric segmentation tasks using the KiTS, MSD Pancreas, and MSD Hepatic Vessel datasets. As shown in Table 1, Rep3D achieves state-of-the-art performance across all settings, consistently outperforming both convolution- and transformer-based baselines. On the KiTS dataset, which includes kidney, tumor, and cyst segmentation, Rep3D achieves the highest average Dice score of 0.736, with strong individual scores of 0.955 (kidney), 0.763 (tumor), and 0.490 (cyst). Notably, Rep3D improves tumor segmentation performance by 2.28% Dice over UNesT-B and 5.39% Dice over 3D UX-Net, demonstrating its ability to adapt to complex local variations in pathological regions. On the MSD Pancreas task, which is particularly challenging due to the pancreas's low contrast and irregular boundaries, Rep3D sets a new benchmark with an average Dice score of 0.723, outperforming SwinUNETR (0.708), nnUNet (0.703), and UNesT-B (0.690). Tumor segmentation also benefits from our reparameterization design, improving by 3.32% Dice compared to 3D UX-Net and 2.03% Dice compared to the fixed-prior variant. On the MSD Hepatic Vessel dataset, Rep3D continues to lead with a mean Dice of 0.674, outperforming the previous best model (UNesT-B, 0.640) and demonstrating superior vessel and tumor localization. The results also highlight the effectiveness of Rep3D's spatially adaptive learning dynamics, especially in sparse and small-structure segmentation where traditional large-kernel convolutions or global self-attention tend to underperform.

*Table 1.* Comparison of SOTA approaches on the three different testing datasets. (*: $p < 0.01$, with Paired Wilcoxon signed-rank test to all baseline networks)

| Methods | #Params | FLOPs | KiTS | | | | MSD Pancreas | | | MSD Hepatic | | |
|---|---|---|---|---|---|---|---|---|---|---|---|---|
| | | | Kidney | Tumor | Cyst | Mean | Pancreas | Tumor | Mean | Hepatic | Tumor | Mean |
| 3D U-Net (Çiçek et al., 2016) | 4.81M | 135.9G | 0.918 | 0.657 | 0.361 | 0.645 | 0.711 | 0.584 | 0.648 | 0.569 | 0.609 | 0.589 |
| SegResNet (Myronenko, 2018) | 1.18M | 15.6G | 0.935 | 0.713 | 0.401 | 0.683 | 0.740 | 0.613 | 0.677 | 0.620 | 0.656 | 0.638 |
| RAP-Net (Lee et al., 2021) | 38.2M | 101.2G | 0.931 | 0.710 | 0.427 | 0.689 | 0.742 | 0.621 | 0.682 | 0.610 | 0.643 | 0.627 |
| nn-UNet (Isensee et al., 2021) | 31.2M | 743.3G | 0.943 | 0.732 | 0.443 | 0.706 | 0.775 | 0.630 | 0.703 | 0.623 | 0.695 | 0.660 |
| TransBTS (Wang et al., 2021) | 31.6M | 110.3G | 0.932 | 0.691 | 0.384 | 0.669 | 0.749 | 0.610 | 0.679 | 0.589 | 0.636 | 0.613 |
| UNETR (Hatamizadeh et al., 2022b) | 92.8M | 82.5G | 0.921 | 0.669 | 0.354 | 0.648 | 0.735 | 0.598 | 0.667 | 0.567 | 0.612 | 0.590 |
| nnFormer (Zhou et al., 2021) | 149.3M | 213.0G | 0.930 | 0.687 | 0.376 | 0.664 | 0.769 | 0.603 | 0.686 | 0.591 | 0.635 | 0.613 |
| SwinUNETR (Hatamizadeh et al., 2022a) | 62.2M | 328.1G | 0.939 | 0.702 | 0.400 | 0.680 | 0.785 | 0.632 | 0.708 | 0.622 | 0.647 | 0.635 |
| 3D UX-Net (k=7) (Lee et al., 2022) | 53.0M | 639.4G | 0.942 | 0.724 | 0.425 | 0.697 | 0.737 | 0.614 | 0.676 | 0.625 | 0.678 | 0.652 |
| UNesT-B (Yu et al., 2023) | 87.2M | 258.4G | 0.943 | 0.746 | 0.451 | 0.710 | 0.778 | 0.601 | 0.690 | 0.611 | 0.645 | 0.640 |
| Rep3D (Fixed Prior) | 65.8M | 757.4G | 0.950 | 0.757 | 0.473 | 0.727 | 0.789 | 0.640 | 0.715 | 0.635 | 0.681 | 0.658 |
| Rep3D | 66.0M | 757.6G | **0.955** | **0.763** | **0.490** | **0.736*** | **0.793** | **0.653** | **0.723*** | **0.650** | **0.697** | **0.674*** |

*Table 2.* Evaluations on the AMOS testing split in different scenarios (*: $p < 0.01$).

| | | | | | | | | | | | | | | | | |
|---|---|---|---|---|---|---|---|---|---|---|---|---|---|---|---|---|
| **AMOS CT (Train From Scratch Scenario)** | | | | | | | | | | | | | | | | |
| Methods | Spleen | R.Kid | L.Kid | Gall. | Eso. | Liver | Stom. | Aorta | IVC | Panc. | RAG | LAG | Duo. | Blad. | Pros. | Avg |
| nn-UNet (350 Ep) | 0.951 | 0.919 | 0.930 | 0.845 | 0.797 | 0.975 | 0.863 | 0.941 | 0.898 | 0.813 | 0.730 | 0.677 | 0.772 | 0.797 | 0.815 | 0.850 |
| nn-UNet (1000 Ep) | 0.967 | 0.958 | 0.945 | 0.890 | 0.818 | 0.979 | 0.914 | 0.953 | 0.920 | 0.824 | 0.799 | 0.743 | 0.823 | 0.900 | 0.867 | 0.887 |
| TransBTS | 0.930 | 0.921 | 0.909 | 0.798 | 0.722 | 0.966 | 0.801 | 0.900 | 0.820 | 0.702 | 0.641 | 0.550 | 0.684 | 0.730 | 0.679 | 0.783 |
| UNETR | 0.925 | 0.923 | 0.903 | 0.777 | 0.701 | 0.964 | 0.759 | 0.887 | 0.821 | 0.687 | 0.688 | 0.543 | 0.629 | 0.710 | 0.707 | 0.740 |
| nnFormer | 0.932 | 0.928 | 0.914 | 0.831 | 0.743 | 0.968 | 0.820 | 0.905 | 0.838 | 0.725 | 0.678 | 0.578 | 0.677 | 0.737 | 0.596 | 0.785 |
| SwinUNETR | 0.956 | 0.957 | 0.949 | 0.891 | 0.820 | 0.978 | 0.880 | 0.939 | 0.894 | 0.818 | 0.800 | 0.730 | 0.803 | 0.849 | 0.819 | 0.871 |
| 3D UX-Net (k=7) | 0.966 | 0.959 | 0.951 | 0.903 | 0.833 | 0.980 | 0.910 | 0.950 | 0.913 | 0.830 | 0.805 | 0.756 | 0.846 | 0.897 | 0.863 | 0.890 |
| 3D UX-Net (k=21) | 0.963 | 0.959 | 0.953 | 0.921 | 0.848 | 0.981 | 0.903 | 0.953 | 0.910 | 0.828 | 0.815 | 0.754 | 0.824 | 0.900 | 0.878 | 0.891 |
| UNesT-B | 0.966 | 0.961 | 0.956 | 0.903 | 0.840 | 0.980 | 0.914 | 0.947 | 0.912 | 0.838 | 0.803 | 0.758 | 0.846 | 0.895 | 0.854 | 0.891 |
| RepOptimizer | 0.968 | 0.964 | 0.953 | 0.903 | 0.857 | 0.981 | 0.915 | 0.950 | 0.915 | 0.826 | 0.802 | 0.756 | 0.813 | 0.906 | 0.867 | 0.892 |
| Rep3D (Fixed) | 0.972 | 0.963 | 0.964 | 0.911 | 0.861 | 0.982 | 0.921 | 0.956 | 0.924 | 0.837 | 0.818 | 0.777 | 0.831 | 0.916 | 0.879 | 0.902 |
| Rep3D (LRBM) | **0.978** | **0.970** | **0.964** | **0.928** | **0.871** | **0.984** | **0.927** | **0.960** | **0.930** | **0.851** | **0.828** | **0.784** | **0.850** | **0.920** | **0.881** | **0.910*** |
| **AMOS MRI (Train From Scratch Scenario)** | | | | | | | | | | | | | | | | |
| Methods | Spleen | R.Kid | L.Kid | Gall. | Eso. | Liver | Stom. | Aorta | IVC | Panc. | RAG | LAG | Duo. | Blad. | Pros. | Avg |
| nn-UNet (350 Ep) | 0.967 | 0.855 | 0.958 | 0.663 | 0.736 | 0.973 | 0.888 | 0.956 | 0.907 | 0.793 | 0.533 | 0.572 | 0.668 | - | - | 0.805 |
| nn-UNet (1000 Ep) | 0.973 | 0.940 | 0.965 | **0.681** | 0.810 | 0.980 | **0.893** | 0.967 | **0.917** | 0.834 | 0.667 | 0.689 | 0.701 | - | - | 0.847 |
| TransBTS | 0.956 | 0.957 | 0.955 | 0.619 | 0.770 | 0.974 | 0.867 | 0.958 | 0.852 | 0.836 | 0.591 | 0.630 | 0.648 | - | - | 0.816 |
| UNETR | 0.942 | 0.956 | 0.930 | 0.552 | 0.741 | 0.967 | 0.836 | 0.947 | 0.829 | 0.815 | 0.564 | 0.621 | 0.624 | - | - | 0.794 |
| nnFormer | 0.949 | 0.952 | 0.950 | 0.601 | 0.758 | 0.972 | 0.859 | 0.960 | 0.843 | 0.832 | 0.569 | 0.618 | 0.637 | - | - | 0.808 |
| SwinUNETR | 0.972 | 0.961 | 0.961 | 0.649 | 0.814 | 0.978 | 0.889 | 0.961 | 0.862 | 0.854 | 0.659 | 0.649 | 0.664 | - | - | 0.836 |
| 3D UX-Net (k=7) | 0.971 | 0.965 | 0.966 | 0.603 | 0.828 | 0.978 | 0.869 | 0.962 | 0.878 | 0.837 | 0.696 | 0.689 | 0.696 | - | - | 0.841 |
| 3D UX-Net (k=21) | 0.968 | 0.962 | 0.967 | 0.610 | 0.830 | 0.977 | 0.858 | 0.954 | 0.880 | 0.829 | 0.701 | 0.697 | 0.700 | - | - | 0.840 |
| UNesT-B | 0.971 | 0.965 | 0.967 | 0.615 | 0.831 | 0.980 | 0.865 | 0.949 | 0.883 | 0.845 | 0.691 | 0.700 | 0.697 | - | - | 0.843 |
| RepOptimizer | 0.970 | 0.967 | 0.971 | 0.635 | 0.823 | 0.978 | 0.875 | 0.963 | 0.882 | 0.850 | 0.689 | 0.691 | 0.711 | - | - | 0.847 |
| Rep3D (Fixed) | 0.972 | 0.965 | 0.970 | 0.644 | 0.838 | 0.980 | 0.883 | 0.965 | 0.893 | 0.861 | 0.714 | 0.701 | 0.725 | - | - | 0.855 |
| Rep3D (LRBM) | **0.975** | **0.969** | **0.975** | 0.657 | **0.845** | **0.984** | 0.891 | **0.970** | 0.901 | **0.879** | **0.718** | **0.721** | **0.750** | - | - | **0.864*** |

## 5.2. Evaluation on Multi-Organ Segmentation

Beyond the ability to segment anatomical structures across scales, we further evaluate Rep3D on the AMOS benchmark under the "train-from-scratch" setting for both CT and MRI modalities. On AMOS-CT, Rep3D achieves the best performance across all 15 evaluated anatomical structures, surpassing strong baselines including SwinUNETR, UNesT, and 3D UX-Net. Notably, Rep3D outperforms UNesT-B by 2.13% and RepOptimizer by 2.02% of average Dice score, while operating with fewer parameters than UNesT. On AMOS-MRI, a more challenging modality due to the variable range of contrast intensity and anatomical ambiguity, Rep3D maintains its superior performance, achieving an average Dice of 0.864, again outperforming all competing approaches. Compared to the best-performing transformer baseline (UNesT-B, 0.854) and convolutional baseline (3D UX-Net (k=21), 0.840), Rep3D delivers consistent improvements across nearly all organ classes, particularly in difficult regions such as the pancreas, gallbladder, and adrenal glands. These gains underscore the effectiveness of our spatially adaptive re-parameterization strategy in enhancing convergence and feature expressivity without increasing model complexity.

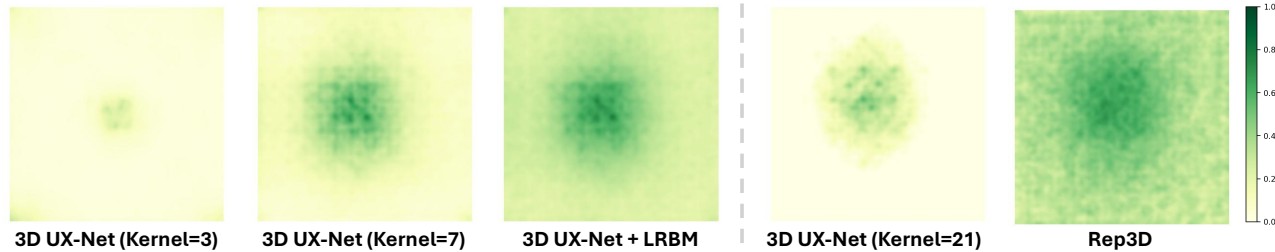

*Figure 3.* As kernel size increases, depthwise convolutions in 3D UX-Net exhibit increasingly diffuse ERFs, gradually expanding the gradient dynamics from local to broader spatial regions. Incorporating LRBM further enhances weighting toward global areas by modulating the spatial contribution of distant elements. In contrast, Rep3D produces a well-distributed ERF that preserves strong central activation while extending contextual influence across the full kernel.

*Table 3.* Ablation Studies on bias generator's convolutional layers and LRBM 3D adaptability with the AMOS testing split.

| Methods | Spleen | R.Kid | L.Kid | Gall. | Eso. | Liver | Stom. | Aorta | IVC | Panc. | RAG | LAG | Duo. | Blad. | Pros. | Avg |
|---|---|---|---|---|---|---|---|---|---|---|---|---|---|---|---|---|
| Kernel=$1 \times 1 \times 1$ | 0.972 | 0.968 | **0.965** | 0.926 | 0.863 | 0.984 | 0.917 | 0.956 | 0.922 | 0.851 | 0.816 | 0.779 | **0.863** | 0.912 | **0.894** | 0.905 |
| Kernel=$3 \times 3 \times 3$ | 0.970 | 0.966 | 0.960 | **0.930** | 0.863 | 0.984 | **0.935** | 0.958 | 0.924 | **0.859** | 0.827 | 0.758 | 0.862 | 0.908 | 0.892 | 0.906 |
| Kernel=$5 \times 5 \times 5$ | 0.974 | 0.967 | 0.964 | 0.925 | 0.833 | 0.984 | 0.924 | 0.956 | 0.910 | 0.850 | **0.829** | **0.786** | 0.843 | **0.921** | 0.884 | 0.903 |
| Kernel=$7 \times 7 \times 7$ | **0.978** | **0.970** | 0.964 | 0.928 | **0.871** | **0.984** | 0.927 | **0.960** | **0.930** | 0.851 | 0.828 | 0.784 | 0.850 | 0.920 | 0.881 | **0.910** |
| 3D UX-Net (k=7) | 0.966 | 0.959 | 0.951 | 0.903 | 0.833 | 0.980 | 0.910 | 0.950 | 0.913 | 0.830 | 0.805 | 0.756 | 0.846 | 0.897 | 0.863 | 0.890 |
| 3D UX-Net + LRBM | 0.968 | 0.963 | 0.952 | 0.911 | 0.841 | 0.981 | 0.915 | 0.959 | 0.920 | 0.835 | 0.811 | 0.770 | 0.851 | 0.901 | 0.872 | 0.897 |

## 5.3. Ablation Studies

**Isolation of Spatial Modulation (LRBM vs. Vanilla).** To strictly isolate the contribution of our proposed Low-Rank Receptive Bias Modeling (LRBM), we compare Rep3D against a "Vanilla Rep" baseline (Appendix Table 7). This baseline utilizes the identical plain encoder architecture (a single $21 \times 21 \times 21$ branch per block) but is trained using standard stochastic gradient descent without spatial modulation. The results are decisive: the Vanilla Rep baseline exhibits significant optimization instability, characterized by high variance in training loss and consistently lower Dice scores. This empirical failure confirms that the "plain" large-kernel architecture is inherently unstable on volumetric data; our learnable spatial prior is not merely an enhancement, but a necessary condition for convergence in this regime.

**Optimal Capacity for Bias Modeling (Generator Depth).** We investigate the representational capacity required to model the spatial gradient prior effectively. By varying the depth of the generator network $f_\theta$ (from 1 to 3 layers), we observe a clear trade-off between expressivity and optimizability. The 1-layer generator proves insufficient, likely because a linear mapping cannot capture the complex, non-isotropic diffusion patterns of the effective receptive field. Conversely, the 3-layer variant introduces optimization difficulties, leading to a slight performance degradation ($0.910 \rightarrow 0.899$ Dice). Based on validation performance, we use the 2-layer configuration as the default design, providing sufficient non-linearity to model nuanced spatial priors while maintaining a lightweight footprint that does not impede gradient flow.

**Local vs. Global Context in Bias Generation (Generator Kernel Size).** While the Rep3D backbone utilizes a massive $21 \times 21 \times 21$ kernel to capture global semantic context, the spatial bias generator may operate on a different scale. We analyzed the generator's kernel size ($K_G \in \{1^3, 3^3, 5^3, 7^3\}$) to determine the optimal receptive field for determining gradient weights. As shown in Table 3, the impact is organ-dependent. Small generator kernels ($1^3$) favor boundary-sensitive organs (e.g., bladder), suggesting that fine-grained local statistics are crucial for edge delineation. However, the $7 \times 7 \times 7$ generator achieves the highest overall Dice (0.910), indicating that a larger receptive field is the best for estimating the spatial prior. This suggests that the decision to up-weight or down-weight a central pixel should be informed by its immediate neighborhood, even if the backbone filter itself spans a much larger volume.

**Universality of LRBM (Integration with 3D UX-Net).** Finally, to demonstrate that our findings are not specific to the Rep3D architecture, we integrate the LRBM module into a standard 3D UX-Net (which uses fixed $7 \times 7 \times 7$ kernels). As reported in Table 3, this integration improves the average Dice score from 0.890 to 0.897. The gains are particularly pronounced in anatomically challenging regions with irregular shapes, such as the left adrenal gland ($0.756 \rightarrow 0.770$) and the duodenum ($0.846 \rightarrow 0.851$). These improvements confirm that spatially adaptive learning rates provide a robust inductive bias that aids convergence in hard-to-segment regions, functioning as a plug-and-play enhancement for existing volumetric segmentation backbones.

*Table 4.* Additional analyses on voxel-spacing sensitivity and inference latency. **(a)** Rep3D remains effective under a finer $1.0 \times 1.0 \times 1.0$ mm spacing, suggesting that the learned spatial re-parameterization is not tied to a single resolution regime. **(b)** Block-level inference latency is measured on an input of size $(1, 1, 96, 96, 96)$. Rep3D provides a larger $21^3$ receptive field while retaining a plain convolutional inference path after re-parameterization.

| (a) Voxel Spacing Sensitivity (Dice ↑) | | | | | |
|---|---|---|---|---|---|
| **Resolution** | **AMOS CT** | **AMOS MRI** | **KiTS** | **Pancreas** | **Hepatic** |
| Default setting | 0.910 | 0.864 | 0.736 | 0.723 | 0.674 |
| $1.0 \times 1.0 \times 1.0$ mm | **0.917** | **0.872** | **0.740** | **0.733** | **0.689** |

| (b) Block-Level Inference Latency (input $1 \times 1 \times 96^3$) | | |
|---|---|---|
| **Block Design** | **Receptive Field / Window** | **Latency (ms)** |
| Conv $3 \times 3 \times 3$ | $3^3$ | $0.743 \pm 0.3$ |
| Swin Transformer | $7^3$ window | $11.256 \pm 0.2$ |
| Rep3D | $21^3$ | $13.092 \pm 0.1$ |

### 5.4. Practical Analysis

**Voxel Spacing Sensitivity and Inference Latency.** As Rep3D uses a distance-decay prior over the 3D kernel support, we first evaluate whether its learned spatial modulation is tied to the voxel spacing used in the main experiments. As shown in Table 4(a), Rep3D remains effective under a finer $1.0 \times 1.0 \times 1.0$ mm resolution and improves Dice across all five benchmarks compared with the default preprocessing setting. This suggests that the proposed re-parameterization is not a narrow scale-specific assumption, but captures a spatial optimization bias that remains useful across resolution regimes. We also report block-level inference latency in Table 4(b). Rep3D is slower than a standard $3^3$ convolution and slightly slower than the measured Swin Transformer block, but directly provides a much larger $21^3$ receptive field while retaining a plain convolutional inference path after re-parameterization. Thus, the comparison should be interpreted as a tradeoff among latency, receptive field size, and architectural simplicity rather than as a pure speed ranking.

**Training-Time Overhead and Deployment Scope.** Rep3D does not eliminate the systems cost of large 3D kernels. Instead, it addresses a complementary optimization bottleneck: naively increasing kernel size often leads to unstable training and limited performance gains. The proposed LRBM module improves the trainability of large kernels during optimization and is removed before inference. Therefore, the deployed model contains no generator or multi-branch structure, and the practical cost of Rep3D is primarily the expected latency and memory footprint of using a large $21^3$ convolutional receptive field.

**Hyperparameter Selection and Transferability.** The LRBM generator depth and kernel size control the capacity of the training-time modulation module. We select the compact 2-layer generator using validation performance, rather than test-set tuning, and use this configuration for

all final test evaluations. We view the observed sensitivity as a capacity tradeoff rather than brittleness: the generator should be expressive enough to refine the distance-decay prior, but lightweight enough to avoid unnecessary optimization complexity. In this paper, the learned modulation is trained jointly with each target dataset and task, so we do not claim direct transferability across anatomies or modalities. Studying whether similar spatial modulation patterns emerge across related 3D tasks is an important future direction.

## 6. Discussion & Limitations

In this work, we introduced Rep3D, a re-parameterization framework that explicitly models spatial convergence dynamics in large kernel 3D convolutions. By linking effective receptive field (ERF) behavior with first-order optimization theory, we demonstrated that large convolution kernels naturally exhibit non-uniform learning dynamics, where central elements converge faster than peripheral ones. To address this, Rep3D integrates a learnable spatial prior via low-rank modulation, allowing the optimizer to differentially emphasize kernel regions with the distinctive characteristics of ERF during training. Our experiments across five diverse 3D segmentation benchmarks, confirm that Rep3D consistently improves performance over both transformer-based and convolution-based SOTA approaches, while maintaining a plain and efficient encoder design. We use direct supervised training rather than external pretrained initialization to isolate the effect of Rep3D as an optimization re-parameterization mechanism. Combining Rep3D with large-scale self-supervised 3D pretraining is left for future work. The success of Rep3D reinforces several broader insights. First, spatially adaptive optimization is a promising direction for bridging inductive biases in CNNs with the dynamic learning capacity of attention-based models. Second, incorporating explicit ERF modeling into kernel design enables more efficient parameter usage, particularly in data-limited medical imaging scenarios. Moreover, our framework enhances network interpretability: the modulation masks can be visualized and aligned with ERF patterns as demonstrated in Figure 3, offering insights into how spatial understanding guides the learning of convolution kernels.

While Rep3D demonstrates strong empirical performance across diverse 3D medical segmentation tasks, several limitations remain. First, although our learnable modulation mechanism introduces minimal architectural overhead, the training cost associated with large 3D kernels (e.g., $21 \times 21 \times 21$) remains nontrivial, particularly in memory-constrained GPU environments. Unlike 2D convolution kernels (i.e. MegEngine packages for 2D depthwise kernels), limited packages and approaches have been proposed to opti-

mize the large kernel mechanism in 3D. This limits the batch size and input resolution during training, which can affect convergence and generalization. Future work could explore progressive training strategies, multi-resolution optimization, or low-resolution proxy supervision to alleviate this constraint while maintaining segmentation fidelity. Second, Rep3D uses a distance-decay prior whose physical interpretation depends on voxel spacing. In our primary experiments, volumes are resampled to standardized resolutions to balance computation and segmentation fidelity. Our additional $1.0 \times 1.0 \times 1.0$ mm experiments suggest that Rep3D remains effective beyond a single spacing regime, indicating that the learned modulation captures a robust spatial optimization bias rather than a narrow resolution-specific prior. Nevertheless, a systematic study across anisotropic acquisitions, heterogeneous scanner protocols, and task-dependent spacing choices remains an important future direction.

## 7. Conclusion

In this paper, we introduced Rep3D, a receptive-biased reparameterization framework for large kernel 3D convolutions. By modeling effective receptive field (ERF) behavior as a learnable spatial prior, Rep3D enables adaptive element-wise learning dynamics during training, bridging the gap between convolutional inductive bias and optimization-aware design. Implemented via a lightweight modulation network, our approach avoids complex multi-branch architectures while improving training efficiency and segmentation accuracy. Extensive experiments across five volumetric medical imaging benchmarks demonstrate consistent improvements over SOTA transformer and CNN approaches, establishing Rep3D as a scalable and effective solution for 3D medical image analysis.

## Impact Statement

This paper studies an optimization phenomenon in large-kernel 3D convolutional networks and proposes a reparameterization technique that improves their training. The contribution is primarily methodological, but we evaluate the method exclusively on medical image segmentation benchmarks. The most direct downstream applications are therefore in clinical imaging workflows, including automated organ and lesion delineation for quantitative assessment, surgical planning, and radiotherapy contouring.

**Positive impact.** More accurate 3D segmentation models have the potential to reduce the manual contouring burden on clinicians and improve the consistency of downstream measurements, such as tumor volume, organ boundaries, and organ-at-risk margins. Rep3D introduces a lightweight modulation module during training, but removes this module at inference. As a result, the deployed model retains a plain

large-kernel convolutional architecture without additional generator or multi-branch inference overhead. This property is important for practical medical imaging systems, where computational resources, memory constraints, and inference simplicity can affect deployability.

**Limitations and risks.** The five benchmarks used in this paper, while public and widely adopted, are drawn from limited data sources and do not reflect the full diversity of scanner vendors, acquisition protocols, institutions, patient demographics, or pathological variants encountered in real-world clinical practice. Improvements in Dice score on held-out test splits do not by themselves establish clinical safety, robustness, or generalization. Models trained with Rep3D should therefore not be deployed for patient care without prospective validation, regulatory clearance where applicable, and human oversight by qualified clinicians. As with all medical AI systems, careful attention to dataset shift, demographic bias, failure detection, and the role of automation in high-stakes clinical decisions remains essential. We release our code openly to support independent reproduction, scrutiny, and further validation.

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

## A. Full Derivation of the Equivalent-Kernel Update

In this section, we provide the full derivation of the spatially non-uniform effective update induced by a two-branch large-kernel re-parameterization block. This derivation complements Sec. 3 and formalizes why the central region of the equivalent large kernel receives stronger optimization signals than the peripheral region.

Consider a two-branch convolutional block with a large kernel $W_L \in \mathbb{R}^{K_L \times K_L \times K_L}$ and a small kernel $W_S \in \mathbb{R}^{K_S \times K_S \times K_S}$, where $K_S < K_L$. The block output is

$$Y = \alpha_L(X * W_L) + \alpha_S(X * W_S), \tag{12}$$

where $X$ denotes the input feature map, $*$ denotes 3D convolution, and $\alpha_L, \alpha_S > 0$ are branch scaling coefficients. Since convolution is linear with respect to the kernel, the two branches can be merged into a single equivalent large kernel:

$$W' = \alpha_L W_L + \alpha_S \operatorname{Pad}(W_S), \tag{13}$$

where $\operatorname{Pad}(\cdot)$ embeds the small kernel into the center of the large-kernel support. Let $\Omega_L$ denote the spatial support of the large kernel and $\Omega_S \subset \Omega_L$ denote the central support covered by the padded small kernel.

During training, the two branches are optimized separately. Let $\mathcal{L}$ be the training loss. For first-order optimization with branch-wise learning rates $\eta_L$ and $\eta_S$, the branch parameters are updated as

$$W_L^{(t+1)} = W_L^{(t)} - \eta_L \frac{\partial \mathcal{L}}{\partial W_L^{(t)}}, \tag{14}$$

$$W_S^{(t+1)} = W_S^{(t)} - \eta_S \frac{\partial \mathcal{L}}{\partial W_S^{(t)}}. \tag{15}$$

The equivalent kernel at iteration $t$ is

$$W'^{(t)} = \alpha_L W_L^{(t)} + \alpha_S \operatorname{Pad}(W_S^{(t)}). \tag{16}$$

Substituting Eq. 14 and Eq. 15 into Eq. 16, the equivalent kernel after one update step becomes

$$\begin{aligned}
W'^{(t+1)} &= \alpha_L W_L^{(t+1)} + \alpha_S \operatorname{Pad}(W_S^{(t+1)}) \\
&= \alpha_L \left( W_L^{(t)} - \eta_L \frac{\partial \mathcal{L}}{\partial W_L^{(t)}} \right) + \alpha_S \operatorname{Pad} \left( W_S^{(t)} - \eta_S \frac{\partial \mathcal{L}}{\partial W_S^{(t)}} \right) \\
&= W'^{(t)} - \alpha_L \eta_L \frac{\partial \mathcal{L}}{\partial W_L^{(t)}} - \alpha_S \eta_S \operatorname{Pad} \left( \frac{\partial \mathcal{L}}{\partial W_S^{(t)}} \right).
\end{aligned} \tag{17}$$

Therefore, the update of the equivalent kernel is

$$\Delta W'^{(t)} = W'^{(t+1)} - W'^{(t)} = -\alpha_L \eta_L \frac{\partial \mathcal{L}}{\partial W_L^{(t)}} - \alpha_S \eta_S \operatorname{Pad} \left( \frac{\partial \mathcal{L}}{\partial W_S^{(t)}} \right). \tag{18}$$

We now examine this update at each spatial offset $\mathbf{p} \in \Omega_L$. For offsets inside the small-kernel support, $\mathbf{p} \in \Omega_S$, both branches contribute to the equivalent-kernel update:

$$\Delta W'^{(t)}(\mathbf{p}) = -\alpha_L \eta_L \frac{\partial \mathcal{L}}{\partial W_L^{(t)}(\mathbf{p})} - \alpha_S \eta_S \frac{\partial \mathcal{L}}{\partial W_S^{(t)}(\mathbf{p})}, \quad \mathbf{p} \in \Omega_S. \tag{19}$$

For offsets outside the small-kernel support, $\mathbf{p} \in \Omega_L \setminus \Omega_S$, only the large branch contributes:

$$\Delta W'^{(t)}(\mathbf{p}) = -\alpha_L \eta_L \frac{\partial \mathcal{L}}{\partial W_L^{(t)}(\mathbf{p})}, \quad \mathbf{p} \in \Omega_L \setminus \Omega_S. \tag{20}$$

Equivalently, the update can be written in a piecewise form:

$$\Delta W'^{(t)}(\mathbf{p}) = \begin{cases} -\alpha_L \eta_L \dfrac{\partial \mathcal{L}}{\partial W_L^{(t)}(\mathbf{p})} - \alpha_S \eta_S \dfrac{\partial \mathcal{L}}{\partial W_S^{(t)}(\mathbf{p})}, & \mathbf{p} \in \Omega_S, \\ -\alpha_L \eta_L \dfrac{\partial \mathcal{L}}{\partial W_L^{(t)}(\mathbf{p})}, & \mathbf{p} \in \Omega_L \setminus \Omega_S. \end{cases} \tag{21}$$

This shows that the central region of the equivalent large kernel receives optimization signals from both the large and small branches, while the peripheral region receives signals only from the large branch.

To make the effective learning-rate interpretation explicit, suppose the two branches receive comparable gradient directions over the overlapping support. Then the center region can be interpreted as having an effective update strength

$$\lambda_{\text{eff}}(\mathbf{p}) = \alpha_L \eta_L + \alpha_S \eta_S, \quad \mathbf{p} \in \Omega_S, \tag{22}$$

whereas the peripheral region has

$$\lambda_{\text{eff}}(\mathbf{p}) = \alpha_L \eta_L, \quad \mathbf{p} \in \Omega_L \setminus \Omega_S. \tag{23}$$

Thus, the equivalent learning strength over the merged large kernel can be summarized as

$$\lambda_{\text{eff}}(\mathbf{p}) = \begin{cases} \alpha_L \eta_L + \alpha_S \eta_S, & \mathbf{p} \in \Omega_S, \\ \alpha_L \eta_L, & \mathbf{p} \in \Omega_L \setminus \Omega_S. \end{cases} \tag{24}$$

This spatially non-uniform effective update is the key observation behind Rep3D. Standard large-kernel re-parameterization implicitly imposes a fixed center-biased optimization pattern determined by the hand-designed small-kernel support $\Omega_S$. In contrast, Rep3D replaces this fixed implicit bias with a learnable spatial modulation field over the full 3D kernel support. This allows the optimization strength of different kernel locations to adapt during training, while the final model remains a plain large-kernel convolution after re-parameterization.

### A.1. Theoretical extension from SGD to Adam/AdamW

We provide a rigorous derivation of how the Constant-Scale Linear Addition (CSLA) block induces a spatially varying learning rate, specifically under adaptive optimization algorithms like Adam (Kingma & Ba, 2014).

**1. The Normalization Cancellation Effect** Consider the update rule for a single parameter $w$ in Adam. The update $\Delta w_t$ at step $t$ is given by:

$$\Delta w_t = -\eta \cdot \frac{m_t}{\sqrt{v_t} + \epsilon}, \tag{25}$$

where $m_t$ is the first moment (momentum) and $v_t$ is the second moment (variance) of the gradients. In a re-parameterized branch scaled by $\alpha$ (i.e., $w_{\text{branch}} = \alpha w$), the gradient is scaled by $\alpha$: $g \leftarrow \alpha g$. Consequently, the moments scale as $m_t \leftarrow \alpha m_t$ and $v_t \leftarrow \alpha^2 v_t$. Substituting these into the update rule:

$$\Delta w_{\text{branch}} = -\eta \cdot \frac{\alpha m_t}{\sqrt{\alpha^2 v_t} + \epsilon} \approx -\eta \cdot \frac{m_t}{\sqrt{v_t}}. \tag{26}$$

As noted, the scaling factor $\alpha$ cancels out. This implies that the magnitude of the update step for any single branch is largely independent of its scaling factor $\alpha$. It is effectively governed only by the global learning rate $\eta$.

**2. Structural Superposition and Effective Step Size** However, such derivation lies in analyzing branches in isolation. The effective kernel $W'$ used during inference is the structural superposition of the branches:

$$W' = \alpha_L W_L + \alpha_S \mathcal{P}(W_S), \tag{27}$$

where $\mathcal{P}(\cdot)$ represents the zero-padding operator that aligns the small kernel to the spatial dimensions of the large kernel.

The effective update $\Delta W'$ is the linear combination of the branch updates:

$$\Delta W'(x) = \alpha_L \Delta W_L(x) + \alpha_S \Delta W_S(x). \tag{28}$$

We can further analyze the magnitude of this effective update across spatial positions $x$:

**Case A: Peripheral Region ($x \in$ Periphery)** These spatial positions exist only in the support of $W_L$. The effective update is:

$$\Delta W'_{\text{peri}} = \alpha_L \Delta W_L \approx \alpha_L \cdot \eta \cdot \vec{u}_L, \tag{29}$$

where $\vec{u}_L$ is the normalized update direction. The step size magnitude is proportional to one unit of update.

**Case B: Central Region ($x \in$ Center)** These spatial positions exist in the support of *both* $W_L$ and $W_S$. The effective update is the superposition of both branches:

$$\Delta W'_{\text{center}} = \alpha_L \Delta W_L + \alpha_S \Delta W_S \approx \alpha_L(\eta \vec{u}_L) + \alpha_S(\eta \vec{u}_S). \tag{30}$$

Assuming the gradient directions $\vec{u}_L$ and $\vec{u}_S$ are not perfectly orthogonal (they are optimizing the same loss on the same features), these vectors constructively interfere. Crucially, even if Adam normalizes $\Delta W_L$ and $\Delta W_S$ to have similar magnitudes, the center receives the sum of two update vectors, while the periphery receives only one.

**3. The Induced Spatial Prior** This derivation proves that structurally re-parameterized blocks inherently induce a spatial learning rate field $\lambda(x)$:

$$\|\Delta W'(x)\| \propto \begin{cases} C_1 \cdot \eta & \text{if } x \in \text{Periphery} \\ (C_1 + C_2) \cdot \eta & \text{if } x \in \text{Center} \end{cases} \tag{31}$$

where $C_1, C_2 > 0$ are constants derived from the structural combination. This confirms that optimization is faster at the center and slower at the periphery, aligning with the "Gaussian" diffusion pattern of Effective Receptive Fields (ERFs), demonstrating similar outcome with the SGD optimizer.

## B. Rep3D Motivations from Experiments: Naively Scaling 3D Convolution Kernels

Recent advances in medical image segmentation have highlighted the importance of large receptive fields for capturing long-range spatial dependencies in volumetric data. Motivated by this, there has been a growing trend toward enlarging convolutional kernel sizes in 3D CNN architectures, such as 3D UX-Net (Lee et al., 2022) and RepUX-Net (Lee et al., 2023), which attempt to mimic the global context modeling capabilities of transformers while retaining the inductive biases of CNNs.

However, the straightforward enlargement of kernel size introduces several practical and theoretical challenges:

- **Optimization Instability:** Large convolutional kernels suffer from slow or uneven convergence, particularly in the outer kernel regions, which are rarely activated in early training. This leads to ineffective utilization of capacity and suboptimal learning behavior.

- **Degrading Performance:** Empirically, simply increasing the kernel size does not guarantee improved performance. Beyond a certain scale, performance tends to saturate or even degrade.

- **Inefficient Parameter Usage:** Naïve kernel scaling introduces a quadratic (especially in 3D) growth in the number of parameters, making training inefficient and difficult to regularize.

To illustrate these effects empirically, we conduct a systematic ablation study on the AMOS dataset using the 3D UX-Net encoder as a backbone. 3D UX-Net is an ideal starting point because it already leverages larger kernels ($7 \times 7 \times 7$) in its

baseline. We vary the convolutional kernel size from $3 \times 3 \times 3$ to $21 \times 21 \times 21$, keeping all other architectural and training settings fixed. The results, shown in Table 4 , confirm our hypothesis.

*Table 5.* Impact of kernel size on segmentation performance using 3D UX-Net encoder on the AMOS dataset.

| Kernel Size | Avg. Dice Score |
|:---:|:---:|
| $3 \times 3 \times 3$ | 0.881 |
| $5 \times 5 \times 5$ | 0.885 |
| $7 \times 7 \times 7$ | 0.890 |
| $9 \times 9 \times 9$ | 0.891 |
| $11 \times 11 \times 11$ | 0.893 |
| $13 \times 13 \times 13$ | 0.894 |
| $15 \times 15 \times 15$ | **0.895** |
| $17 \times 17 \times 17$ | 0.893 |
| $19 \times 19 \times 19$ | 0.893 |
| $21 \times 21 \times 21$ | 0.891 |

As seen in Table 4, performance initially improves with increasing kernel size, peaking at $15 \times 15 \times 15$. However, further enlargements yield diminishing or even negative returns, despite increased computational cost. These findings reveal a key limitation of naïve kernel enlargement: Although it increases theoretical receptive field, it fails to translate into meaningful gains due to the lack of spatially adaptive optimization.

## C. Data Preprocessing & Training Details

*Table 6.* Hyperparameters for direction training scenario on four public datasets

| Hyperparameters | Direct Training |
|:---|:---:|
| Encoder Stage | 4 |
| Layer-wise Channel | $48, 96, 192, 384$ |
| Hidden Dimensions | 768 |
| Patch Size | $96 \times 96 \times 96$ |
| No. of Sub-volumes Cropped | 2 |
| Training Steps | 60000 |
| Batch Size | 1 |
| AdamW $\epsilon$ | $1e-8$ |
| AdamW $\beta$ | $(0.9, 0.999)$ |
| Peak Learning Rate | $1e-4$ |
| Learning Rate Scheduler | ReduceLROnPlateau |
| Factor & Patience | 0.9, 10 |
| Dropout | X |
| Weight Decay | 0.08 |
| Data Augmentation | Intensity Shift, Rotation, Scaling |
| Cropped Foreground | ✓ |
| Intensity Offset | 0.1 |
| Rotation Degree | $-30°$ to $+30°$ |
| Scaling Factor | x: 0.1, y: 0.1, z: 0.1 |

We apply hierarchical steps for data preprocessing: 1) intensity clipping is applied to further enhance the contrast of soft tissue

(AMOS CT, KiTS, MSD Pancreas:{min:-175, max:250}; MSD Hepatic Vessel:{min:0, max:230}); AMOS MRI:{min:0, max:1000}. 2) After clipping, we perform intensity normalization for each volume using min-max normalization: $(X - X_1)/(X_{99} - X_1)$ to normalize the intensity value between 0 and 1, where $X_p$ denote as the $p_{th}$ percentile of intensity in $X$. We then perform downsampling to certain voxel spacing (i.e. AMOS CT, MSD hepatic vessels, MSD Pancreas and KiTS: $1.5 \times 1.5 \times 2.0$, AMOS MRI: $1.0 \times 1.0 \times 1.0$) randomly crop sub-volumes with size $96 \times 96 \times 96$ at the foreground and perform data augmentations, including rotations, intensity shifting, and scaling (scaling factor: 0.1). All training processes with Rep3D are optimized with either Stochastic Gradient Descent (SGD) or AdamW optimizer. We trained all models for 60000 steps using a learning rate of 0.0001 on an NVIDIA A100 GPU across all datasets. One epoch takes approximately about 9 minutes for KiTS, 5 minutes for MSD Pancreas, 12 minutes for MSD hepatic vessels, 7 minutes for AMOS CT and 1 minute for AMOS MRI, respectively.

All experiments are conducted under a direct supervised learning setting. For the KiTS and MSD datasets, we employ a 5-fold cross-validation strategy using an 80%/10%/10% split for training, validation, and testing, respectively. For the AMOS dataset, we use a fixed single split with the same partitioning ratio. Details on training procedures and preprocessing protocols are provided in the supplementary material. Our proposed re-parameterization approach Rep3D, is benchmarked against both convolutional and transformer-based state-of-the-art (SOTA) methods for 3D medical image segmentation. For nnUNet (Isensee et al., 2021), we evaluate performance across two different training schedules to account for fairness, since Rep3D is trained with 60,000 iterations (approximately equivalent to 350 epochs). Therefore, we have provided performance with partial scheduled (350 epochs) and full scheduled (1000 epochs) to demonstrate our model generalizability.

### C.1. Datasets Details

We leverage four challenging public datasets across different scales: 1) AMOS22 (MICCAI 2022 Abdominal Multi-organ Segmentation Challenge) (Ji et al., 2022): Comprises 200 multi-contrast abdominal CT scans with 15 organ-level anatomical labels and 33 MRI scans with 13 organ-level anatomical labels for comprehensive abdominal segmentation, 2) KiTS21 (MICCAI 2021 Kidney Tumor Segmentation Challenge) (Heller et al., 2019): Includes 210 contrast-enhanced abdominal CT scans from the University of Minnesota Medical Center (2010–2018), with manual annotations for kidney, tumor, and cyst, 3) MSD Pancreas (Medical Segmentation Decathlon) (Antonelli et al., 2022): Contains 282 abdominal contrast-enhanced CT scans annotated for both pancreas and pancreatic tumor segmentation, and 4) MSD Hepatic Vessel (Medical Segmentation Decathlon) (Antonelli et al., 2022): Contains 303 abdominal CT scans annotated for hepatic vessel and associated tumor segmentation.

*Table 7.* Complete overview of Four public datasets

| Challenge | AMOS CT | AMOS MR | MSD Pancreas | MSD Hepatic Vessels | KiTS |
|---|---|---|---|---|---|
| Imaging Modality | Multi-Contrast CT | Multi-Contrast MRI | Venous CT | | Arterial CT |
| Anatomical Region | Abdomen | | Pancreas | Liver | Kidney |
| Sample Size | 200 | 33 | 282 | 303 | 300 |
| Anatomical Label | Spleen, Left & Right Kidney, Gall Bladder, Esophagus, Liver, Stomach, Aorta, Inferior Vena Cava (IVC) Pancreas, Left & Right Adrenal Gland (AG), Duodenum Bladder (CT only), Prostate/Uterus (CT only) | | Pancreas, Tumor | Hepatic Vessels, Tumor | Kidney, Tumor |
| Data Splits | 1-Fold (Internal) | | | 5-Fold Cross-Validation | |
| | Train: 160 / Validation: 20 / Test: 20 | Train: 22 / Validation: 4/ Test: 7 | Train: 225 / Validation: 27 / Testing: 30 | Training: 242, Validation: 30 / Testing: 31 | Training: 240, Validation: 30 / Testing: 30 |
| 5-Fold Ensembling | N/A | N/A | X | ✓ | X |

### C.2. Network Architecture

We adopt a 3D encoder-decoder architecture from both 3D UX-Net (Lee et al., 2022) and SwinUNETR (Hatamizadeh et al., 2022a) as the backbone of Rep3D. Instead of using encoder block with feed forward layer, we simply using a plain convolutional design with depthwise separable convolutions in parallel with LRBM. The encoder consists of 4 hierarchical stages with increasing feature dimensions and depthwise convolutions of large kernel size ($21 \times 21 \times 21$), followed by a symmetric decoder for volumetric segmentation. The encoder includes:

- An initial input projection block with a $7 \times 7 \times 7$ convolution (stride 2, padding 3) followed by a residual block with two $3 \times 3 \times 3$ convolutions and GELU activations.

- Stage 1: 2 Rep3D blocks with 48 channels followed by a strided $2 \times 2 \times 2$ convolution for downsampling.

- Stage 2: 2 Rep3D blocks with 96 channels, followed by a strided $2 \times 2 \times 2$ convolution for downsampling.

- Stage 3: 2 Rep3D blocks with 192 channels, followed by a strided $2 \times 2 \times 2$ convolution for downsampling.

- Stage 4: 2 Rep3D blocks with 384 channels, followed by a strided $2 \times 2 \times 2$ convolution for downsampling.

Each stage modulates large kernel weights using a learnable re-parameterization mask computed via a lightweight 2-layer generator network within each Rep3D block. For each Rep3D block, it includes:

- A single depthwise 3D convolution with a large kernel size of $21 \times 21 \times 21$ and padding size of 10, followed by a layer normalization and a GELU activation.

- A 2-stage lightweight generator network including:
    - First layer: a depthwise $7 \times 7 \times 7$ convolution followed by layer normalization and a sigmoid activation.
    - Second layer: another depthwise $7 \times 7 \times 7$ convolution followed by layer normalization.

The decoder mirrors the encoder and consists of:

- 4 upsampling modules (UnetrUpBlock from MONAI), each with a transpose convolution (stride 2), skip connection, and a residual block with two $3 \times 3 \times 3$ convolutions and GELU activations.

- 1 output projection block (UnetOutBlock from MONAI) consisting of a $1 \times 1 \times 1$ convolution to map to the number of target classes.

## D. Training Efficiency Comparison

To further validate our claims around improved convergence behavior and training efficiency, we conducted an additional ablation study focusing on runtime and performance dynamics across different configurations of Rep3D on the AMOS dataset. While our primary goal is to improve segmentation accuracy and spatial convergence, it is equally important that such gains are achieved with minimal training overhead. In this study, we compare three architectural variants:

(a) **Vanilla Rep:** Rep3D with parallel convolutional branches, but without any spatial prior modulation.

(b) **Fixed Prior:** Rep3D with a non-learnable reciprocal distance mask acting as a fixed spatial prior.

(c) **Full LRBM:** Rep3D with a learnable low-rank bias module (LRBM), modulating the spatial prior adaptively via a generator network.

All models were trained under identical conditions: using a single NVIDIA A100 GPU, batch size of 2, and AdamW optimizer. We report the validation Dice scores at key training checkpoints (10k, 20k, 40k, and 60k iterations), as well as the total training time to convergence.

*Table 8.* Training efficiency and convergence comparison of different Rep3D variants on AMOS dataset.

| Method | Time (hrs) | 10k Iter | 20k Iter | 40k Iter | 60k Iter |
|---|---|---|---|---|---|
| Vanilla Rep | 17.3 | 0.853 | 0.868 | 0.886 | 0.892 |
| Fixed Prior | 15.5 | 0.864 | 0.875 | 0.892 | 0.902 |
| Full LRBM | 17.5 | **0.871** | **0.885** | **0.897** | **0.910** |

As observed in Table 8, the full LRBM variant consistently achieves the best segmentation accuracy at every checkpoint, demonstrating accelerated convergence. The introduction of the learnable low-rank generator yields a modest increase in training time (+0.2 hours compared to Vanilla Rep), but this is substantially outweighed by the observed performance gains. The Fixed Prior variant also performs competitively, suggesting the benefit of incorporating even a static spatial prior.

**D.1. Validation Experiments on Variable Branch Learning rate**

*Table 9.* Quantitative Evaluation on Variable Learning Rates in Parallel Branches

| Optimizer | Main Branch | Para. Branch | Train Steps | Main LR | Para. LR | Mean Dice |
|-----------|-------------|--------------|-------------|---------|----------|-----------|
| SGD | $21 \times 21 \times 21$ | $\times$ | 60000 | 0.0005 | $\times$ | 0.849 |
| SGD | $21 \times 21 \times 21$ | $\times$ | 60000 | 0.0004 | $\times$ | 0.852 |
| SGD | $21 \times 21 \times 21$ | $\times$ | 60000 | 0.0003 | $\times$ | 0.856 |
| SGD | $21 \times 21 \times 21$ | $\times$ | 60000 | 0.0002 | $\times$ | 0.859 |
| SGD | $21 \times 21 \times 21$ | $\times$ | 60000 | 0.0001 | $\times$ | 0.854 |
| AdamW | $21 \times 21 \times 21$ | $\times$ | 60000 | 0.0005 | $\times$ | 0.855 |
| AdamW | $21 \times 21 \times 21$ | $\times$ | 60000 | 0.0004 | $\times$ | 0.859 |
| AdamW | $21 \times 21 \times 21$ | $\times$ | 60000 | 0.0003 | $\times$ | 0.861 |
| AdamW | $21 \times 21 \times 21$ | $\times$ | 60000 | 0.0002 | $\times$ | 0.862 |
| AdamW | $21 \times 21 \times 21$ | $\times$ | 60000 | 0.0001 | $\times$ | 0.860 |
| SGD | $21 \times 21 \times 21$ | $3 \times 3 \times 3$ | 60000 | 0.0002 | 0.0006 | 0.872 |
| SGD | $21 \times 21 \times 21$ | $3 \times 3 \times 3$ | 60000 | 0.0002 | 0.0005 | 0.869 |
| SGD | $21 \times 21 \times 21$ | $3 \times 3 \times 3$ | 60000 | 0.0002 | 0.0004 | 0.867 |
| SGD | $21 \times 21 \times 21$ | $3 \times 3 \times 3$ | 60000 | 0.0002 | 0.0003 | 0.870 |
| SGD | $21 \times 21 \times 21$ | $3 \times 3 \times 3$ | 60000 | 0.0002 | 0.0001 | 0.865 |
| AdamW | $21 \times 21 \times 21$ | $3 \times 3 \times 3$ | 60000 | 0.0002 | 0.0006 | 0.887 |
| AdamW | $21 \times 21 \times 21$ | $3 \times 3 \times 3$ | 60000 | 0.0002 | 0.0005 | 0.886 |
| AdamW | $21 \times 21 \times 21$ | $3 \times 3 \times 3$ | 60000 | 0.0002 | 0.0004 | 0.887 |
| AdamW | $21 \times 21 \times 21$ | $3 \times 3 \times 3$ | 60000 | 0.0002 | 0.0003 | 0.889 |
| AdamW | $21 \times 21 \times 21$ | $3 \times 3 \times 3$ | 60000 | 0.0002 | 0.0001 | 0.886 |

To empirically validate the theoretical insight of the spatially varying convergence dynamics in parallel-branched re-parameterization, we initially perform experiments using the CSLA block with Rep3D network architecture, composing of a main large kernel branch ($21 \times 21 \times 21$) and a parallel small kernel branch ($3 \times 3 \times 3$), with separate learning rates applied to each. As shown in Table 3, the single-branch design (no parallel branch) performance improved moderately with lower learning rates with both SGD and AdamW. The Dice score peaks at 0.859 with a learning rate of 0.0002 using SGD, and AdamW achieves its best performance of 0.862 at 0.0002 as well. However, with the addition of a small kernel parallel branch and using a higher learning rate for the small kernel (e.g., $\lambda_S > \lambda_L$), we observed consistent improvements across all configurations. Specifically, the best result with SGD reached 0.872 when using $\lambda_L = 0.0002$ and $\lambda_S = 0.0006$. Similarly, AdamW attained a maximum Dice score of 0.889 with $\lambda_L = 0.0002$ and $\lambda_S = 0.0003$. These results validate our hypothesis that assigning higher learning rates to the small kernel branch accelerates convergence of central kernel regions, while maintaining stability in peripheral regions with a lower learning rate for the large kernel. Moreover, such results further confirm that spatially varying convergence behavior can be approximated through differentiated learning rates, supporting the design principle behind our learnable re-parameterization in Rep3D.

# E. Ablation Study on Network Depth for LRBM

*Table 10.* Ablation Study on Network Depth for LRBM with the AMOS testing split

| Number of Layers | Spleen | R. Kid | L. Kid | Gall. | Eso. | Liver | Stom. | Aorta | IVC | Panc. | RAG | LAG | Duo. | Blad. | Pros. | Avg |
|------------------|--------|--------|--------|-------|------|-------|-------|-------|-----|-------|-----|-----|------|-------|-------|-----|
| 1 Layer | 0.974 | 0.965 | 0.964 | 0.925 | 0.859 | 0.982 | 0.926 | 0.956 | 0.920 | 0.842 | 0.824 | 0.781 | 0.842 | 0.915 | 0.879 | 0.904 |
| 2 Layers | **0.978** | **0.970** | 0.964 | **0.928** | **0.871** | **0.984** | **0.927** | **0.960** | **0.930** | **0.851** | **0.828** | **0.784** | **0.850** | **0.920** | **0.881** | **0.910** |
| 3 Layers | 0.971 | 0.964 | **0.965** | 0.924 | 0.841 | 0.983 | 0.920 | 0.952 | 0.910 | 0.839 | 0.819 | 0.779 | 0.837 | 0.910 | 0.870 | 0.899 |

**E.1. Additional Comparisons with nnU-Net Variants: ResEnc nnU-Net, STU-Net, MedNeXt**

we conducted further comparisons with recent state-of-the-art (SOTA) 3D medical image segmentation architectures, including **ResEnc nnU-Net** (Isensee et al., 2021), **STU-Net-H** (Huang et al., 2023), and **MedNeXt** (Roy et al., 2023). These models have demonstrated strong performance across various benchmarks and provide important context for positioning Rep3D among contemporary architectures. All baseline methods were trained under their official recommended training schedules (typically 1000 epochs), using the same computational resources and training splits for fair comparison.

*Table 11.* Average Dice scores across five segmentation benchmarks under full training schedules.

| Method | AMOS CT | AMOS MRI | KiTS | Pancreas | Hepatic |
|---|---|---|---|---|---|
| nnU-Net (1000 epochs) | 0.887 | 0.847 | 0.706 | 0.703 | 0.660 |
| ResEnc nnU-Net | 0.892 | 0.850 | 0.711 | 0.706 | 0.661 |
| STU-Net-H | 0.900 | 0.848 | 0.707 | 0.712 | 0.648 |
| MedNeXt | 0.897 | 0.856 | 0.720 | 0.713 | 0.663 |
| U-Mamba | 0.904 | 0.859 | 0.729 | 0.716 | 0.665 |
| **Rep3D (Ours)** | **0.910** | **0.864** | **0.736** | **0.723** | **0.674** |

These comparisons further validate the strong and consistent performance of **Rep3D** across multiple challenging 3D segmentation benchmarks. Even under extensive training schedules (1000 epochs), Rep3D outperforms the SOTA alternatives, demonstrating its robustness and generalizability.

## F. Impact Statement

This paper presents Rep3D, a spatially-adaptive re-parameterization framework for improving the optimization of large-kernel 3D convolutional networks. The contribution is primarily methodological: we study non-uniform convergence behavior in large 3D kernels and introduce a lightweight learnable spatial bias that improves training stability and segmentation performance. Because our experiments are conducted on medical image segmentation benchmarks, the most direct downstream applications are in clinical imaging workflows, including organ and lesion delineation, quantitative measurement, surgical planning, and radiotherapy contouring.

The potential positive impact of this work is to improve the accuracy and reliability of 3D medical image segmentation models while preserving a simple inference-time architecture. More accurate segmentation can reduce the manual annotation burden on clinicians and improve the consistency of measurements such as organ boundaries, tumor volume, and organ-at-risk margins. In addition, Rep3D's modulation module is used only during training and removed at inference time, so the deployed model does not require an additional generator or multi-branch inference structure. This property may be useful in practical medical imaging environments where memory, latency, and implementation simplicity are important constraints.

At the same time, this work should not be interpreted as demonstrating clinical readiness. The datasets used in this study, while public and widely adopted, do not capture the full diversity of scanner vendors, acquisition protocols, institutions, patient populations, anatomical variation, or pathological presentations encountered in real-world practice. Improvements in Dice score on held-out benchmark splits do not by themselves establish clinical safety, robustness, fairness, or generalization under distribution shift. Models trained with Rep3D should therefore not be deployed for patient care without rigorous external validation, prospective evaluation, regulatory review where applicable, and appropriate human oversight by qualified clinical experts.

Potential risks include over-reliance on automated segmentation outputs, failure under unseen imaging protocols or rare pathologies, and biased performance across underrepresented patient groups or acquisition settings. Future work should evaluate Rep3D under multi-institutional deployment scenarios, anisotropic and heterogeneous voxel-spacing regimes, clinically meaningful failure modes, and demographic subgroup analyses. We release the source code to support independent reproduction, scrutiny, and further validation by the research community.

