# OpenReview forum: "Spatially-Adaptive Gradient Re-parameterization for 3D Large Kernel Optimization"
_ICML.cc/2026/Conference — ICML 2026 regular_

### Official Review · Reviewer_6Y13 · 2026-02-18

**Soundness:** 3
**Presentation:** 2
**Significance:** 3
**Originality:** 3
**Overall Recommendation:** 3
**Confidence:** 4

**Summary:**

This paper presents Rep3D, a framework aimed at improving the stability and effectiveness of large-kernel 3D convolutions for medical image segmentation. The authors argue that simply increasing kernel sizes often results in optimization difficulties and diminishing performance gains. The paper offers a theoretical perspective on why structural re-parameterization strategies tend to work. In essence, multi-branch designs implicitly introduce spatially varying learning dynamics, where central kernel regions receive stronger gradient updates than peripheral ones. Building on this observation, the authors introduce a Low-Rank Receptive Bias Modeling (LRBM) module. Rather than relying on computationally expensive parallel branches, Rep3D employs a lightweight generator network to produce a learnable scaling mask. Across five volumetric segmentation datasets, Rep3D demonstrates consistent improvements over both transformer-based approaches and CNN-based baselines.

**Compliance With Llm Reviewing Policy:**

Affirmed.

**Key Questions For Authors:**

1. How does the inference latency of a 21×21×21 Rep3D block compare with (a) a standard 3×3×3 convolution block and (b) a window-based transformer block such as Swin?
2. Is the learned spatial modulation mask dataset-specific, or could it be transferred across tasks or anatomies to accelerate training?

**Limitations:**

1. Training with large kernels demands significant GPU memory, with batch sizes reportedly limited even on high-end hardware.
2. The approach currently relies on controlled downsampling strategies. Its behavior at finer voxel spacings remains less clear.

**Strengths And Weaknesses:**

- Strengths:
1. A notable contribution is the theoretical analysis connecting structural re-parameterization with gradient behavior. The authors show that “large + small” parallel branches can be interpreted as a single operator with spatially non-uniform learning rates, effectively explaining why the kernel center tends to converge faster.
2. Rep3D replaces heavy multi-branch training structures with a compact generator that produces a modulation mask. This design captures spatial bias effects while avoiding the memory and computational overhead typically associated with parallel convolution branches.
3. The experimental section is thorough, covering multiple challenging datasets. Rep3D achieves the best Dice scores across all five benchmarks

- Weaknesses:
1. Although the multi-branch overhead is removed, large 3D kernels are inherently expensive. The reported experiments indicate substantial GPU memory usage, limiting batch sizes and potentially constraining practical deployment.
2. The effectiveness of the spatial prior appears dependent on specific voxel spacings. Performance saturation at higher resolutions suggests that the distance-decay assumption may not generalize seamlessly across scales.
3. Performance seems tied to particular generator settings (e.g., depth and kernel size). Deviations lead to noticeable drops, indicating some dependence on careful hyperparameter tuning.

---

> ### Author Rebuttal · Authors · 2026-03-28
>
> We thank the reviewer for the thoughtful summary and for recognizing both the theoretical motivation and the consistent empirical gains of Rep3D across the five benchmarks.
>
> **Concern 1: On training cost and practical deployment.**
> We agree that large 3D kernels are computationally demanding. That said, the practical challenge is not only memory or runtime, but also that **simply increasing kernel size often does not pay off**. The optimization becomes unstable and performance can saturate or degrade. Rep3D addresses this complementary bottleneck. Rather than claiming to eliminate the systems cost of large-kernel 3D convolutions, it provides an initial practical route for making larger receptive fields actually effective, through a lightweight training-time re-parameterization that improves optimization while keeping the deployed inference architecture plain and efficient.
>
> **Concern 2: On voxel spacing / scale sensitivity.**
> We thank the reviewer for this important question. To test whether Rep3D is tied to a particular voxel-spacing regime, we performed additional experiments at a finer 1.0×1.0×1.0 resolution. Rep3D continues to outperform the results with current resolution (i.e 1.5x1.5x1.5) as follows:
> | Resolutions | AMOS CT | AMOS MRI | KiTS | Pancreas | Hepatic
> |---|---|---|---|---|---|
> | 1.5x1.5x1.5 (Current Paper Setting) | 0.910 | 0.864 | 0.736 | 0.723 | 0.674
> | 1.0x1.0x1.0 | **0.917** | **0.872** | **0.740** | **0.733** | **0.689**
>
> Suggesting that the proposed re-parameterization generalizes beyond the specific spacing used in the main paper. More broadly, this indicates that the distance-decay prior serves as a robust spatial inductive bias, while the lightweight generator learns an adaptive refinement of this bias during optimization. We therefore do not view the prior as a narrow scale-specific assumption, but as a learnable receptive bias that remains effective across different resolution regimes.
>
> **Concern 3: On hyperparameter sensitivity.**
> We agree that generator depth and kernel size affect performance. This is a natural consequence of the proposed design, in which the generator must have enough capacity to adaptively refine the distance-decay prior into an effective spatial modulation pattern for the downstream optimization dynamics, while remaining lightweight enough to avoid unnecessary complexity. We therefore interpret this not as brittleness, but as the expected tradeoff in choosing the capacity of a training-time re-parameterization module. The purpose of the ablation is to identify a compact and robust default configuration, and the selected 2-layer generator should be understood in this sense rather than as a highly fragile optimum.
>
> **Concern 4: Regarding inference latency.**
> Regarding inference latency, we now provide a quantitative block-level comparison on an input of (1, 1, 96, 96, 96). As expected, a standard 3×3×3 convolution block is the fastest (0.743 ms), reflecting its small kernel and limited local receptive field. A Swin Transformer block also shows somewhat lower latency than a Rep3D 21×21×21 block (11.246 ms vs. 13.092 ms), which is reasonable given its smaller 7×7×7 local window. In contrast, Rep3D directly provides a much larger 21×21×21 receptive field while retaining a plain convolutional inference path, since the lightweight generator is removed after re-parameterization. We therefore view this comparison not as a pure speed ranking, but as a tradeoff between latency, receptive field size, and architectural simplicity. For clarity, we provide the quantitative comparison table below and will include it in the appendix.
>
> | Block Design | Latency (ms)
> |---|---|
> | Conv 3x3x3 | 0.743 $\pm$ 0.3
> | Swin Transformer | 11.256  $\pm$ 0.2
> | Rep3D | 13.092 $\pm$ 0.1
>
> **Concern 5: Regarding transferability of the learned mask.**
> We thank the reviewer for this insightful question. In the current paper, the learned spatial modulation mask is trained jointly with the target dataset and downstream task, so we do not claim transferability in the present submission. However, since the mask is intended to encode a structured spatial optimization bias rather than task-specific semantic content, it is plausible that parts of the learned re-parameterization may be shared across related anatomies or segmentation settings. We believe this is a promising future direction. Studying the learned modulation/weight distributions across tasks could help determine whether transferable spatial bias patterns exist and whether they can be reused to improve optimization efficiency.

---

### Official Review · Reviewer_nfp2 · 2026-02-20

**Soundness:** 3
**Presentation:** 3
**Significance:** 3
**Originality:** 3
**Overall Recommendation:** 5
**Confidence:** 4

**Summary:**

In their work, the authors present Rep3D, a framework for re-parametrization using spatial bias in large convolutional kernels. A generator is used to generate learnable, distance-decaying kernels during training. The authors explain this idea based on the observation, that when large and small kernels are combined, learning rates differ between periphery and center. The approach is evaluated on five 3D medical image segmentation benchmarks. State-of-the-art performance is claimed thorughout these datasets, tested against common CNN and Transformer baselines.

**Compliance With Llm Reviewing Policy:**

Affirmed.

**Final Justification:**

The rebuttal confirmed my already high score for this work.

**Key Questions For Authors:**

See weakness

**Limitations:**

yes

**Strengths And Weaknesses:**

**Strength**:

1) The authors strongly motivate their approach based on theory and observations of previous work. The mathematical explanations are sound and give clear value to the manuscript.
2) The proposed approach shows increased performance while imporving efficiency during inference. Especially in the domain of 3D segmentation, this is highly desirable.
3) The validation over five different datasets and a variety of compared methods is strong and legitimates the authors claims.
4) The ablation studies are extensive, give value to the paper and allow further insights into the proposed method.
5) The paper is well written and structured.

**Weakness**:

1) While the efficiency during inference is evaluated clearly and shows improvement against the compared methods, the authors do not evaluate the training cost of their approach. Metrics showing the memory footprint and wall-time during training in comparison to the compared methods would be desirable. Equal memory consumption and wall-time would strengthen the approach even further.
2) While the evaluation is rigorous with 5 different datasets, they all are medical datasets. A non-medical dataset would further strengthen this work and truly allow the claim of SOTA on multiscale-datsets. Otherwise the claim should be changed accordingly.

---

> ### Author Rebuttal · Authors · 2026-03-28
>
> We sincerely thank the reviewer for the positive assessment of our work, for recognizing the theoretical motivation of Rep3D, and for highlighting the strength of the empirical validation and ablation studies.
>
> **Concern 1: On training cost.**
> We thank the reviewer for this helpful suggestion. We agree that training-time cost is an important practical consideration, especially for large 3D kernels. In fact, we already include a training-efficiency comparison in the appendix on AMOS under identical settings (**single NVIDIA A100, batch size 2, AdamW**), comparing **Vanilla Rep, Fixed Prior, and Full LRBM**. The key result is that the added LRBM module introduces only minimal training overhead relative to the vanilla large-kernel baseline (**17.5h vs. 17.3h**), while consistently improving convergence and achieving the best validation Dice at every checkpoint (**0.871/0.885/0.897/0.910 vs. 0.853/0.868/0.886/0.892 for Vanilla Rep**). In addition, all variants were trained with the same hardware budget and batch size, suggesting that the lightweight LRBM does not introduce a prohibitive additional memory burden relative to the vanilla large-kernel baseline, rather the dominant memory cost remains the large 21×21×21 3D kernel itself. These results suggest that Rep3D improves optimization quality with only a very modest additional training cost and without materially changing the practical memory regime. We agree that making this training-side evidence more visible would strengthen the paper, and we will highlight these results more explicitly in the revision.
>
> **Concern 2: On scope of evaluation.**
> We agree that the empirical scope of this paper is 3D medical segmentation, and our claim should be read accordingly. The results support state-of-the-art performance on the evaluated 3D medical segmentation benchmarks, rather than a universal claim over all non-medical or multiscale datasets. We will revise the wording to make this scope explicit and agree that extension to other 3D domains is a valuable future direction.

---

> > ### Author Rebuttal · Reviewer_nfp2 · 2026-04-01
> >
> > Thank you for the rebuttal.
> > The wall-clock time comparison (17.5h vs. 17.3h) demonstrates that the explicit gradient modulation and the LRBM module introduce negligible overhead. I appreciate you agreeing to scope down your claims slightly.
> >
> > Given that my primary concerns have been fully addressed, I am maintaining my score of Accept.

---

> > > ### Author Response · Authors · 2026-04-04
> > >
> > > We sincerely thank Reviewer nfp2 for the encouraging and thoughtful feedback. We greatly appreciate your positive assessment of the paper and your recognition of both the theoretical motivation and the empirical validation. Your comments are very valuable to us, and we will incorporate your suggestions to further improve the final manuscript.

---

### Official Review · Reviewer_GiQr · 2026-03-10

**Soundness:** 2
**Presentation:** 2
**Significance:** 2
**Originality:** 3
**Overall Recommendation:** 4
**Confidence:** 5

**Summary:**

This paper addresses the optimization challenges associated with scaling up 3D convolution kernels for high-resolution volumetric medical image segmentation.Motivated by the observation that structurally re-parameterized convolutions implicitly induce spatially varying learning rates that mirror the natural diffusion patterns of Effective Receptive Fields (ERFs), the authors propose Rep3D. Rep3D utilizes a  "Low-Rank Receptive Bias Modeling" (LRBM) mechanism. Rep3D employs a lightweight generator network to produce a receptive-biased modulation mask conditioned on a distance decay prior, encouraging faster convergence in the central local regions.

**Compliance With Llm Reviewing Policy:**

Affirmed.

**Final Justification:**

The authors have addressed concerns I raised, I am raising the score.

**Key Questions For Authors:**

1) Can the authors justify why hyperparameter and architectural tuning (such as the depth of the LRBM generator) was conducted on the AMOS testing split, despite having a designated validation set?
2) Can the authors provide benchmark comparisons against more contemporary 3D medical image segmentation models?
3) Can the authors significantly condense or remove the useless mathematical derivations in Section 3.1 (Equations 1-10) in a revised version? The authors can directly report the results of ablation study for the claims in this section.

**Limitations:**

yes

**Strengths And Weaknesses:**

Strengths:
1) The paper identifies a crucial optimization bottleneck in scaling 3D convolution kernels. The proposed method is simple yet effective, and easy to follow. The proposed distance decay prior matches the physical intuition of how ERFs diffuse.
2) The evaluation contains five different datasets and demonstrates SOTA performance.

Weaknesses:
1) Math-washing. The paper contains unnecessary equations in Section 3.1. Equations (1) through (10) merely represent trivial applications of the basic calculus chain rule. Those equations actually did not prove the empirical claims. The actual conclusion stems entirely from the empirical ablation studies, making the extensive math-washing and the excessive space devoted to it in the main text completely unnecessary.
2) Outdated baselines. The paper claims to achieve state-of-the-art (SOTA) performance across multiple 3D volumetric datasets. However, the evaluated baselines are all back to 2023 or earlier. Authors fail to benchmark Rep3D against recent advances in 3D medical image segmentation from 2024 or 2025 (e.g. U-Mamba).
3) Data leakage in model selection. The authors explicitly report conducting design ablations on the test set. Table 9 is titled "Ablation Study on Network Depth for LRBM with the AMOS testing split" , and Section 5.3 uses these test-set results to justify selecting the 2-layer generator as the optimal design. Appendix A.2 states that the AMOS dataset used a partitioned split , and Table 6 explicitly lists the AMOS CT data split as "Train: 160 / Validation: 20 / Test: 20". Tuning architectural hyperparameters on the testing split is kind of data leakage and undermines the credibility of the paper's state-of-the-art claims.
4) As defined in Eq. (14) and Eq. (15), the generator $f_\theta$ strictly takes the deterministic distance mapping $P$ as its only input. However $P$ does not condition on the dynamic input feature map or image data. The resulting mask $M$ is merely a static learned tensor, contradicts the core claim of dynamic adaptability to semantic variations. The authors can simply optimize $M$ directly as a learnable parameter matrix rather than using a generator.

---

> ### Author Rebuttal · Authors · 2026-03-28
>
> We thank the reviewer for recognizing the importance of the optimization bottleneck in scaling 3D convolution kernels and for noting the empirical strength of the results across five datasets.
>
> **Concern 1: The role of the derivation on Section 3.1**
> We thank the reviewer for this feedback. We agree that the derivation in Sec. 3.1 is not intended to serve as a stand-alone proof of all empirical improvements. Its purpose is more modest but central to the paper: to make explicit that the commonly used “large-kernel + small-kernel” structural re-parameterization induces spatially non-uniform effective updates across kernel locations, with stronger optimization emphasis near the center than the periphery. This observation is **the theoretical motivation for replacing an implicit fixed bias with the learnable spatial modulation used in Rep3D**. We agree that the current presentation may allocate too much main-text space to the algebraic steps, and in a revision we would be happy to condense the derivation and move lower-level details to the appendix, while keeping the key mechanistic insight in the main paper.
>
> **Concern 2: On comparisons with recent baselines.**
> We appreciate the reviewer’s request for stronger contemporary baselines. In addition to the appendix comparisons against ResEnc nnU-Net, STU-Net-H, and MedNeXt, we further benchmarked U-Mamba using the same preprocessing, splits, and evaluation protocol. Rep3D remains superior on all five benchmarks, improving over U-Mamba from 0.904→0.910 on AMOS CT, 0.859→0.864 on AMOS MRI, 0.725→0.736 on KiTS, 0.716→0.723 on Pancreas, and 0.665→0.674 on Hepatic. This suggests that the advantage of Rep3D is not limited to older baselines, but persists against more recent 3D medical segmentation models.
>
> | Method | AMOS CT | AMOS MRI | KiTS | Pancreas | Hepatic
> |---|---|---|---|---|---|
> | ResEnc nnU-Net | 0.892 | 0.850 | 0.711 | 0.706 | 0.661
> | STU-Net-H | 0.900 | 0.848 | 0.707 | 0.712 | 0.648
> | MedNeXt | 0.897 | 0.856 | 0.720 | 0.713 | 0.663
> | U-Mamba | 0.904 | 0. 859 | 0. 729 | 0.716 | 0.665
> | **Rep3D (Ours)** | **0.910** | **0.864** | **0.736** | **0.723** | **0.674**
>
> **Concern 3: On the AMOS split / model selection concern.**
> We thank the reviewer for pointing this out. We would like to clarify that **our experimental pipeline follows the standard train/validation/test protocol**. Architectural ablations and checkpoint/model selection are conducted using **the validation split**, and **the test split is used only once for final held-out evaluation**. Specifically, table 9 is intended as a held-out comparison of fixed design variants, not as a tuning procedure. The current wording in Table 9 and Section 5.3 is unfortunately ambiguous and may incorrectly suggest that the 2-layer generator was selected on the AMOS test split. This is not the case, and we will revise the caption and surrounding text to remove the ambiguity. We therefore respectfully disagree that the paper suffers from data leakage or that this issue undermines the credibility of the reported SOTA results.
>
> **Concern 4: On whether the modulation mask is “dynamic.”**
> We thank the reviewer for this important point. We agree that the proposed modulation is not dynamic **in a per-sample feature-conditioned sense**, since the current LRBM does not take the input image/feature map as conditioning. Our intended meaning of “adaptive” is **optimization-time spatial adaptivity**, in which the model learns how different kernel locations should be weighted during training, instead of relying on a fixed manually specified spatial prior. We will revise the wording to avoid ambiguity. Moreover, we intentionally use a lightweight generator rather than directly learning an unconstrained mask tensor, because the generator provides a structured and low-rank ERF-aligned parameterization, encouraging regularity, locality, and smoother spatial bias than a free parameter matrix.

---

> > ### Author Rebuttal · Reviewer_GiQr · 2026-04-03
> >
> > Thanks for the authors' detailed response to my concerns. I am raising my score accordingly. I would like to see your revisions. Especially removing the meaningless derivation and clarifying for your model selection.

---

> > > ### Author Response · Authors · 2026-04-04
> > >
> > > We sincerely thank Reviewer GiQr for the careful follow-up and for engaging with our rebuttal in detail. We truly appreciate your updated assessment with the raised score and your openness to our clarifications. Your feedback has been very helpful in improving both the presentation and the positioning of the paper, and we will make sure these improvements are reflected in the revision.

---

### Official Review · Reviewer_7ea8 · 2026-03-13

**Soundness:** 3
**Presentation:** 3
**Significance:** 2
**Originality:** 2
**Overall Recommendation:** 3
**Confidence:** 3

**Summary:**

This paper finds the key problem that naive expansion of convolution kernel size easily leads to optimization instability and performance saturation in high-resolution 3D volumetric data analysis. Based on the inherent spatial bias characteristics of the effective receptive field (ERF), it theoretically proves that structural reparameterization blocks induce spatially differentiated learning rates, which are crucial for convergence. Accordingly, a 3D large-kernel convolution optimization framework called Rep3D is proposed. This framework generates a receptive-biased scaling masks through a lightweight modulation network and performs spatially adaptive re-weighting of the convolutional kernel gradient updates during training, integrating spatial inductive bias with an optimization-aware learning mechanism, without additional computational overhead during the inference phase. Rep3D achieves state-of-the-art performance in five challenging 3D medical segmentation benchmarks, consistently outperforming baseline models based on Transformer and traditional CNN.

**Compliance With Llm Reviewing Policy:**

Affirmed.

**Final Justification:**

Due to concerns of novelty, I decide to keep my score.

**Key Questions For Authors:**

See the weaknesses.

**Limitations:**

yes

**Strengths And Weaknesses:**

Strengths:
1. Previous large-kernel convolutions primarily relied on the structural reparameterization of multi-branch structures (e.g., CSLA) for stable training. This paper theoretically prove that this parallel structure of "large kernel + small kernel" essentially applies a non-uniform learning rate across the spatial dimension (faster convergence in the central region and slower convergence in the peripheral region), providing a solid theoretical basis for introducing a distance decay prior based on the effective receptive field (ERF).
2. The proposed LRBM module dynamically generates a spatial modulation mask through a lightweight generator to adjust the gradient updates of the large kernel during training. This mask is applied only during training, and the generator can be removed during inference, leading to no additional memory or computational burden during inference.
3. The experimental design is rigorous. In the main experiments, five 3D medical segmentation benchmark datasets are selected for testing and comparison. Simultaneously, a variety of ablation experiments are conducted to assess module necessity, hyperparameter sensitivity, and generalization.
4. The paper is easy to follow.
Weaknesses:
1. The paper lacks innovation. The core idea of ​​gradient reparameterization comes from RepOptimizer, the combination of structural reparameterization and large-kernel convolution comes from RepLKNet, and the theoretical basis of spatial prior comes from the classic ERF theory. The core improvement in this paper is only to replace the fixed gradient scaling factor in RepOptimizer with an ERF-based spatial adaptive mask, and to adapt it for 3D medical scenarios.
2. Although inference is lightweight, training large kernel convolutions directly in 3D space incurs extremely high training memory overhead. As the authors acknowledge in the limitations section, the lack of efficient low-level implementations for 3D large kernels severely limits the batch size and input volume resolution during training. This is a fatal weakness in real-world medical imaging research scenarios where computational resources are limited.
3. The models are all training from scratch. However, in current medical image analysis research, using pre-trained backbone(e.g. self-supervised models) on large-scale datasets has become the standard paradigm. Will the strategies introduced in this model seamlessly integrate with pre-trained models to achieve better performance, or will they disrupt the original feature distribution of the pre-trained models?
4. Each figure and Table should be referenced in the main text of the paper. Please include a reference to Figure 1 in the paper.

---

> ### Author Rebuttal · Authors · 2026-03-28
>
> We sincerely thank the reviewer for the positive summary of our work, for recognizing the theoretical motivation behind Rep3D, and for noting the rigor of the experiments and ablations.
>
> **Concern 1: On novelty.**
> We agree that Rep3D builds on several important prior directions, including structural re-parameterization, large-kernel CNNs, and ERF analysis. Our intended claim is not that we introduce these ingredients individually, but that we provide **a new optimization formulation that connects them**. Specifically, our paper shows that the commonly used “large-kernel + small-kernel” structural re-parameterization implicitly induces **spatially non-uniform convergence across kernel elements**, and then replaces this implicit fixed behavior with **an explicit learnable spatial modulation field in 3D**. In this sense, Rep3D is not simply “RepOptimizer with an ERF mask” (i.e. RepOptimizer applies learnable scaling to updates), whereas our method derives **a spatially structured, ERF-inspired, element-wise modulation mechanism for large 3D kernels**, motivated by the convergence asymmetry induced by structural re-parameterization. This theoretical-to-method bridge is the main novelty of the paper.
>
> **Concern 2: On training memory cost.**
> We agree that training large 3D kernels is computationally demanding. However, this is a general systems challenge of the large-kernel 3D regime, rather than a weakness introduced specifically by Rep3D. In practice, the issue is not only memory cost, and is more about **naively enlarging kernel size often fails to translate into better performance and can even degrade optimization stability**, making large receptive fields difficult to use effectively in real 3D segmentation settings. We view Rep3D as an important **initial step toward practical deployment of larger 3D kernels**, instead of adding heavy multi-branch structures. It introduces a lightweight training-time re-parameterization that makes enlarged kernels train more reliably while preserving a plain inference-time architecture. Thus, our contribution is not to claim that large 3D kernels are universally cheap, but to make this otherwise unstable regime meaningfully usable without additional inference-time complexity.
>
> **Concern 3: On pretrained backbones.**
> We agree that evaluating with pretrained backbones is important. At present, however, in 3D medical image analysis, pretrained backbones are more mature and widely adopted on the transformer side (e.g., SwinUNETR, STU-Net), whereas pretrained 3D convolutional backbones remain less standardized. For this reason, training from scratch is still a common and fair protocol for isolating architectural or optimization improvements in 3D segmentation. We chose this setup specifically to evaluate the effect of Rep3D without confounding gains from external pretraining. More importantly, Rep3D is **a training-time spatial re-parameterization** rather than a permanent inference-time architectural modification. The lightweight generator is removed after training, and the deployed model remains a plain convolutional network. For this reason, we do not expect Rep3D to fundamentally disrupt a pretrained feature space. Rather, it should in principle be compatible with pretrained initialization and downstream fine-tuning, while modulating the optimization dynamics of large kernels during adaptation. We agree that a systematic study combining Rep3D with large-scale self-supervised or pretrained backbones is an important future direction, and we will clarify this scope in the revision.
>
> **Minor Concern:**
> Thank you for catching the missing reference to Figure 1. We will add an explicit reference to Figure 1 in the main text and carefully ensure that each figure/table is referenced.

---

> > ### Author Rebuttal · Reviewer_7ea8 · 2026-04-04
> >
> > Thanks for the rebuttal. Due to concerns of novelty, I decide to keep my score. While I am equipped to evaluate the general algorithmic methodology, its specific application to medical image analysis is not my core specialty (e.g. the related works). Thus, more weight should be placed on the other reviewers’ evaluations for fairness.

---

> > > ### Author Response · Authors · 2026-04-04
> > >
> > > We sincerely thank Reviewer 7ea8 for the careful follow-up and for taking the time to read our rebuttal. We truly appreciate your thoughtful and balanced response, and we are especially grateful for your transparency regarding the limits of your domain-specific expertise. Your feedback has still been very valuable in helping us sharpen the novelty positioning and improve the presentation of the paper, and we will make these revisions carefully in the final version.

---

### Decision · Program_Chairs · 2026-04-30

**Decision:**

Accept (regular)

**Comment:**

The rebuttal addressed the concerns raised by the reviewers and provided comprehensive experiments and analysis of the proposed method. The rebuttal and authors’ feedback were helpful for assessing the paper's contributions. The reviewers recommend two accept/weak accept after discussion, and the AC concur. The final version should include all reviewer comments, suggestions, and additional experiments from the rebuttal.